**Technology**

# Empowering integrative and collaborative exploration of single-cell and spatial multimodal data with SGS genome browser

## Graphical abstract

## Authors

Tingting Xia (夏婷婷), Jiahe Sun (孙家贺), Yongjiang Luo (罗永江), ..., Ling Xu (徐凌), Fang Lu (鲁方), Yi Wang (王翊)

## Correspondence

lufang0411@sina.com (F.L.), yiwang28@swu.edu.cn (Y.W.)

## In brief

SGS is a user-friendly, collaborative, and versatile browser for visualizing single-cell and spatial multiomics data. With advanced features for comparative visualization, multi-panel coordinate view, abundant visualization functions, and collaborative exploration, SGS empowers researchers to unlock novel insights from scMulti-omics data.

## Highlights

- SGS is a user-friendly tool for scMulti-omics data visualization and collaboration

- Enhances the visualization of genome-mapped data and multi-panel coordination

- SGS enables cross-modal, spatial tissues and features visualization comparisons

- SGS archives the 3D transcriptomic data visualization with surface model plots

 Xia et al., 2025, Cell Genomics 5, 100848
May 14, 2025 © 2025 The Author(s). Published by Elsevier Inc.

# Cell Genomics

CellPress

## Technology

# Empowering integrative and collaborative exploration of single-cell and spatial multimodal data with SGS genome browser

Tingting Xia (夏婷婷),[1,3] Jiahe Sun (孙家贺),[1,3] Yongjiang Luo (罗永江),[1] Hailong Guo (郭海龙),[1] Yudi Mao (毛禹狄),[1] Ling Xu (徐凌),[2] Fang Lu (鲁方),[1,*] and Yi Wang (王翊)[1,4,*]

[1]Integrative Science Center of Germplasm Creation in Western China (CHONGQING) Science City, Biological Science Research Center, Southwest University, Chongqing, China
[2]State Key Laboratory of Plant Environmental Resilience, College of Biological Sciences, China Agricultural University, Beijing, China
[3]These authors contributed equally
[4]Lead contact
*Correspondence: lufang0411@sina.com (F.L.), yiwang28@swu.edu.cn (Y.W.)

## SUMMARY

Recent advancements in single-cell and spatial omics technologies have generated a large amount of complex, high-dimensional data, which poses significant challenges to visualization tools. We introduce SGS (single-cell and spatial genomics system), a user-friendly, collaborative, and versatile browser designed for visualizing single-cell and spatial multimodal data. SGS excels in the integrative visualization of complex multimodal data, offering an innovative genome browser, flexible visualization modes, and 3D spatially resolved transcriptomics (SRT) data visualization capabilities. Notably, SGS empowers users with advanced capabilities for comparative visualization through features like scCompare, scMultiView, and the dual-chromosome mode. It supports a variety of data formats and is compatible with established analysis tools, enabling collaborative data exploration and visualization without programming. Overall, SGS is a comprehensive browser that enables researchers to unlock novel insights from multimodal data.

## INTRODUCTION

Single-cell and spatial multimodal omics (scMulti-omics) technologies have empowered the measurement of diverse modalities,[1,2] including RNA expression, protein abundance, DNA methylation, chromatin accessibility, and both two- and three-dimensional (2D and 3D) spatial information at the cellular level.[3–5] scMulti-omics has revolutionized our comprehension of gene regulation, epigenetic variations, protein expression dynamics, and cellular interactions, providing profound insights into complex biological processes.[6,7] Driven by innovative platforms, diverse tools, and novel methodologies, single-cell and spatial omics research has undergone unprecedented growth in recent years, generating vast amounts of multi-omics data.[8,9] These multiomics data exhibit substantial complexity,[10] characterized by the simultaneous presence of diverse and distinct feature sets within individual cells, such as (1) transcriptome and chromatin accessibility,[11] (2) transcriptome and DNA methylation,[12] and (3) transcriptome and histone modifications.[13,14] Moreover, multimodal data have inherently abundant feature dimensions, spanning a wide range from hundreds of features for protein epitopes to hundreds of thousands for chromatin-accessible sites.[15] To achieve comprehensive visualization and extract valuable insights from these datasets, the integration of information across multiple omics levels, comparative visualization, as well as enabling efficient information retrieval is needed. How to jointly visualize and collaboratively explore these multiomics data of such high dimensionality, high noise, and diversity poses significant challenges.

Many visualization tools have been developed to address these challenges.[16,17] Tools such as Cellxgene,[18] UCSC Cell Browser,[19] Loupe Browser, ST Viewer,[20] and TissUUmaps3[21] enable interactive exploration of single-cell features in lower-dimensional uniform manifold approximation and projection (UMAP) or t-distributed stochastic neighbor embedding (t-SNE) spaces, along with categorical, spatial dimensions and feature annotation. However, these visualization tools are often designed for specific modalities or data types, such as scRNA or single-cell assay for transposase accessible chromatin (scATAC), which limits their ability to handle the growing complexity of multimodal data and perform comparative visualization effectively. Researchers are often forced to utilize disconnected visualization tools to explore various aspects of multimodal data, which hinders multimodal data integration and visualization, reduces the efficiency of data visualization, and increases the difficulty of data management. Several tools have been developed to bridge the gap in visualizing more complex multimodal data, including AtlasXplore[22] and Vitessce.[23] However, these tools have notable constraints in effectively visualizing complex epigenomic multimodal data (single-cell expression quantitative trait loci [sc-eQTL], single-cell methylation and transcriptome sequencing [scM&T-seq],[24] Droplet Hi-C,[25] and

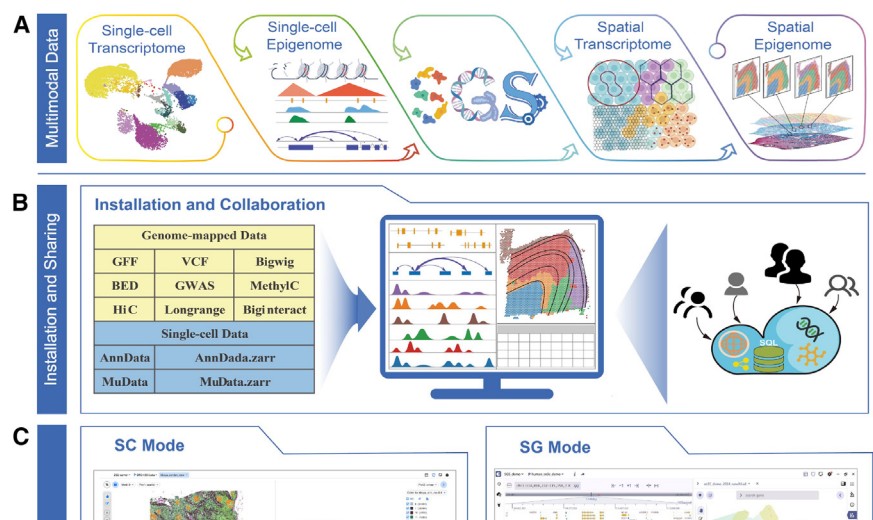

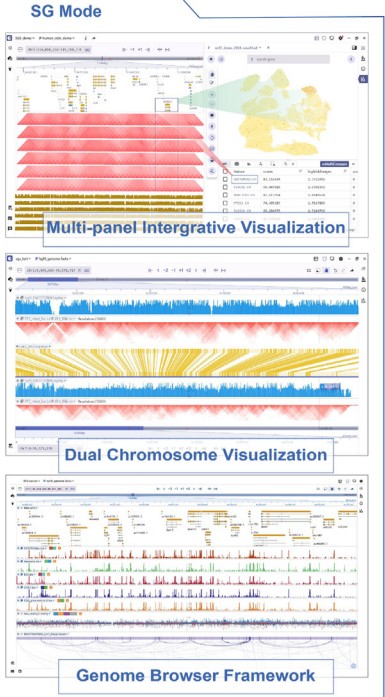

**Figure 1. Overview of SGS**

(A) SGS is a visualization tool designed for single-cell and spatial multimodal data.

(B) SGS supports various data formats, including AnnData and MuData and genome-mapped files (GFF, VCF, BED, HiC, MethylC, GWAS).

(C) SGS offers two visualization modes: SC mode and SG mode. SC mode features 2D or 3D embedding visualization, scCompare, and scMultiView. SG mode includes multi-panel integrative visualization, dual-chromosome visualization, and a novel genome browser framework.

See also Figures S1 and S4–S8 and Table S1.

and spatial sample comparisons. Furthermore, we introduce SGS's newly developed genome browser, which enhances the visualization of genome-mapped data, featuring a dual chromosome mode and a multi-panel interactive design for comprehensive exploration of complex epigenomic multimodal data. We applied SGS to diverse scMulti-omics datasets, including human dorsal prefrontal cortex (PFC) and hippocampus (HPC) sn-m3C-seq data, mouse brain 10x Genomics Visium data, human OneK1K sc-eQTL data, mouse spatial-ATAC-seq data, and Drosophila 3D SRT data. These applications demonstrate SGS's potential in visualizing multimodal data and facilitating the systematic interpretation of cellular heterogeneity, tissue organization, and biological processes.

## DESIGN

SGS is a graphical user interface (GUI)-based visualization tool designed for the integrative and collaborative visualization of high-dimensional single-cell and spatial omics data. Developed using Docker technology and leveraging the Flask and Flutter frameworks, SGS adopts a front-end and back-end separation strategy. It utilizes memory mapping, data chunking, and caching techniques to achieve efficient data parsing and rendering. Additionally, we have designed an adaptive communication mechanism to enhance coordinated visualization between the single-cell and genome browser panels (STAR Methods).

## RESULTS

### Overview of SGS and its advantages over other tools

SGS provides a unified interface for visualizing scMulti-omics data, including genomics, transcriptomics, proteomics, and epigenomics data, at both single-cell and spatial resolutions (Figure 1A). Compared to existing integrated visualization tools, especially Vitessce, SGS has core advantages in terms of

single-nucleus methyl-3C sequencing [sn-m3C-seq][26]), 3D spatially resolved transcriptomics (SRT) data visualization, comparative visualization, usability, and collaborative exploration. This highlights the urgent need for the development of new visualization tools. Such tools should be more accessible, versatile, and capable of facilitating in-depth interpretation of scMulti-omics data within a unified framework.

To address these limitations, we developed the single-cell and spatial genomics system (SGS) as an innovative solution for enhancing the exploration of scMulti-omics data. We introduce the distinctive features and advantages of SGS, emphasizing its user-friendliness and collaborative capabilities in multimodal data visualization and management. We detail how SGS supports interactive visualization of single-cell clustering, spatial tissue sections, and 3D spatial transcriptomic data, and how it leverages functions such as scCompare and scMultiView to provide rich plotting capabilities for comprehensive cross-modal

visualization capabilities, interactivity, view coordination, multi-user collaboration, and user-friendliness (for a detailed comparison, see Figures S1–S3 and Table S1). The key advantages of SGS follow. (1) Enhanced accessibility and collaboration: SGS aims to empower research teams in swiftly establishing a visualization and sharing platform for collaborative data exploration, co-annotation, comments, and project management without programming (Figure 1B). (2) Multimodal integration and coordinate visualization: SGS offers two modes: SC mode and SG mode, with adaptable interface layouts and advanced capabilities (Figure 1C). Notably, SGS significantly enhances the visualization of epigenomic modalities, including scATAC, scMethylC, sc-eQTL, and scHiC, through a novel genome browser framework. To synchronously explore multimodal datasets with linked views across different panels, an adaptive communication mechanism was used to bridge visualization components. These features effectively address the limitations of existing visualization tools in the integration and visualization of complex epigenomic multimodal data. (3) 3D SRT data visualization: SGS outperforms existing visualization tools like Vitessce by offering an interactive and shareable 3D SRT data browser. This allows users to interactively browse 3D data with surface model plots and explore 3D gene expression heterogeneity. (4) Comparative visualization of different modalities and features: SGS offers scCompare, scMultiView, and dual-chromosome mode functions to facilitate comparative visualization of cellular heterogeneity, gene expression, and genome-mapped signals of multimodal data. (5) Compatibility with diverse data formats and tools: SGS supports various data formats, including AnnData, MuData, Zarr, and genome-mapped files (GFF, VCF, BED, HiC, Biginteract, Longrange, MethylC, GWAS [genome-wide association study], Bedgraph). It provides the SgsAnnData R package for format conversion, which is compatible with analysis tools such as Seurat,[27] ArchR,[28] Signac,[29] and Giotto[30] (Figure S4).

## Main features
### Enhanced user-friendliness via graphical installation and interface ease

Efficient big data visualization is essential for research teams. However, building a customized data visualization platform can be challenging and time-consuming for non-technical users. While existing mainstream integrated visualization tools like Vitessce allow users to customize their visualization platforms, they require programming expertise. Vitessce relies on complex configuration options and specific input data formats to tailor view components, which require users to understand detailed configuration parameters and significantly increases the learning curve (Figure S5A). In contrast, SGS is a highly user-friendly, cross-platform GUI visualization software that offers rich graphical operation features (Figure S5B). This provides a fast and user-friendly visualization platform for non-programmers, emphasizing team collaboration and real-time data sharing. SGS leverages Docker and Flutter technologies to achieve graphical one-click installation, avoiding the need for complex software configuration and web service deployment. SGS overcomes the challenges of complex configurations and environment dependencies while ensuring compatibility across Linux, Windows, and MacOS platforms. Users can perform operations through the graphical interface to complete installation, theme settings, single-cell panel, and genome track style settings without the need to edit configuration files directly (Figure S6). After installation, SGS supports batch uploading of various data formats, including GFF, VCF, BigWig, AnnData, and MuData, enabling the rapid visualization of large datasets. Overall, SGS excels in user-friendliness, making advanced data visualization accessible to users with diverse technical backgrounds, ultimately empowering researchers to focus more on their scientific inquiries than their technical challenges.

### Multi-user collaborative data visualization and management

Team collaboration is a much-needed feature in the exploration of scMulti-omics data visualization, and SGS significantly outperforms existing visualization tools such as Vitessce in this critical area. SGS provides a robust framework for collaborative data exploration through its advanced features, as in the following. (1) Real-time collaboration: multiple users can collaborate on the same datasets, with features like cell-type annotation, genome feature renaming, document commenting, and synchronization. Leveraging these functions, multiple users can collaboratively visualize genome-mapped and single-cell data (Figure S7). (2) Seamless data sharing: SGS allows users to share view sessions or web URLs, which makes it easy for team members to access and collaborate on data visualizations. (3) Advanced project and user management: SGS is designed for the needs of multi-project and multi-species research, featuring a robust data management system (Figure S8A). It empowers users to manage multiple projects effectively, offering capabilities for batch addition, deletion, and data grouping to enhance efficiency and flexibility. In addition, SGS allows adding multiple users and assigning different permissions, enabling seamless collaboration in a shared visualization environment and data management (Figure S8B). In summary, SGS has brought significant enhancement to research teams by improving collaboration features and data and user management efficiency, making it a valuable tool for scientific research teams aiming to efficiently visualize and share large datasets.

### Single-cell and spatial omics data visualization
#### SC visualization mode

The SC mode is specifically designed for the comparative exploration of expression-based scMulti-omics data, such as scRNA, spatial transcriptomics (ST), and scProteomics. The SC mode supports embedding plots (generated from UMAP or t-SNE) and visualization of tissue slices (Figure 2A). Users can overlay various annotation metadata on the plots, such as cell type and sample details (Figure 2B). Additionally, users can adjust point size, transparency, the brightness of the tissue image and perform zooming, dragging, and selecting regions of interest. At the bottom of the main interface, essential information is prominently displayed, including tissue slices, meta chart, marker feature table, metadata table, and cell subset (Figures 2C–2E). These features enable the examination of cell annotations, the exploration of marker expression, the comparison of cell compositions, and the visualization of multiple spatial slices.

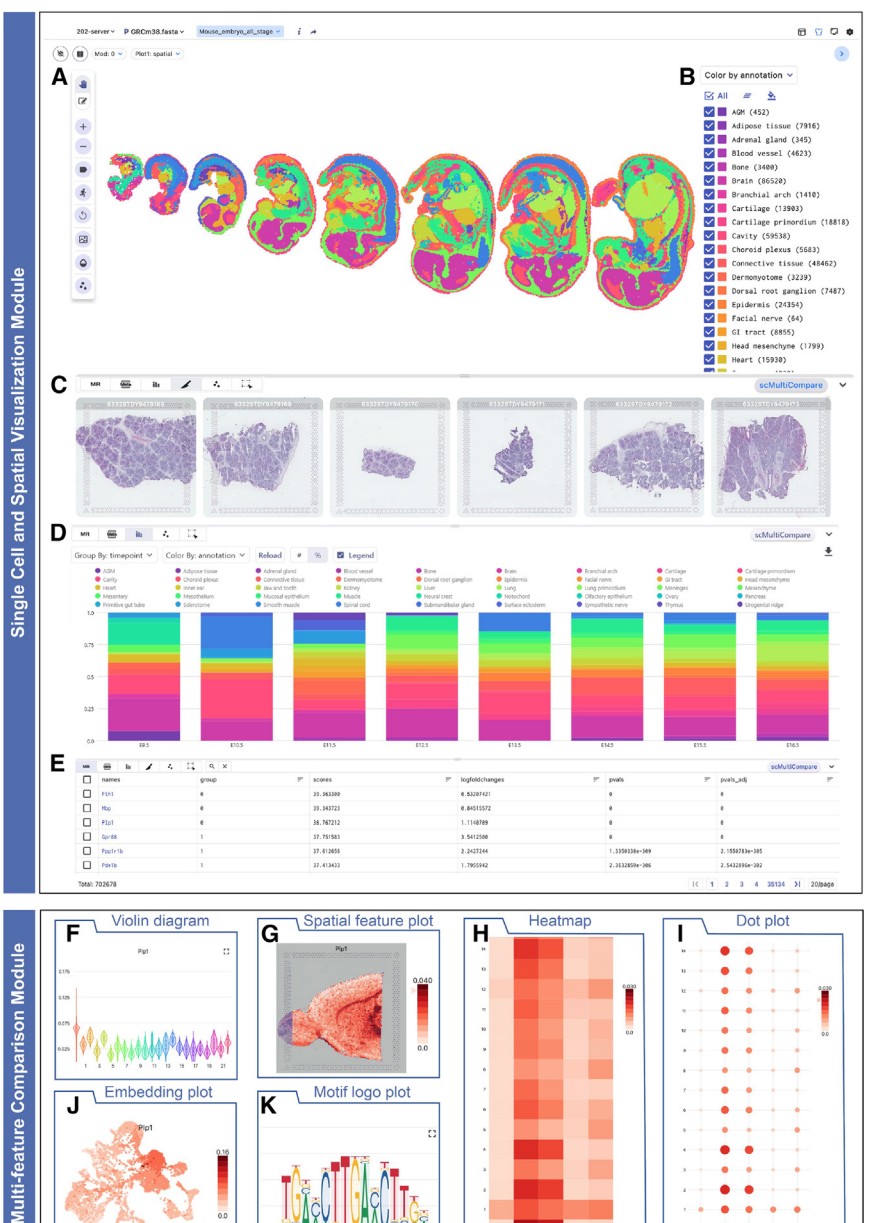

**Figure 2. Key visualization features of SC mode**

(A and B) Embedding plot (A) and annotation information (B) in the main panel of the SC mode.

(C) The spatial distribution of tissue slices from multi-sample datasets, displayed in the Spatial Slice tab.

(D) The cell proportion presented in the Meta Chart tab.

(E) Marker gene information, including gene name, group, and $p$ value.

(F–K) Visualization options of the scMultiView mode, featuring violin plot (F), spatial feature plot (G), heatmap (H), dot plot (I), embedding plot (J), and motif logo plot (K).

See also Figure S9.

Additionally, SGS offers 3D transcriptomic data visualization capabilities that enhance the understanding of gene expression patterns within a 3D context, providing deeper insights into spatial relationships. Users can engage in interactive 3D data exploration with surface model plots, visualizing each gene's spatial expression pattern through color opacity that reflects expression strength. Furthermore, users can adjust the transparency of the mesh models and the size and transparency of points, animate the mesh models, and examine the expression heterogeneity of genes in the 3D view. This real-time, interactive 3D data visualization accelerates studies related to processes such as embryogenesis and organogenesis (Figure S9).

In general, the SC visualization mode offers a dynamic interface for exploring high-dimensional datasets. Its core features include single-cell embedding plots, metadata visualization, 3D SRT visualization, and comparative visualization of multiple spatial samples or genes. These functionalities are helpful in identifying differentially expressed genes, annotating cell types, exploring cellular heterogeneity, and understanding complex biological processes at the single-cell level.

### SG visualization mode

We have developed the SG visualization mode to address the complexities of visualizing single-cell and spatial epigenomic omics data by integrating the genome browser framework with single-cell components. To increase the compatibility of genome-mapped data and enhance the scalability, customizability, and interactivity of the SG visualization mode, we have developed a novel, flexible, and scalable genome browser framework. This framework enables a wide range of track visualizations, including gene structures, genomic variation loci,

In various research studies, the comparative visualization function is essential. The scCompare function of SGS enables users to easily compare specific cell-type annotations and gene expressions across modalities (e.g., peak, gene score, motif assays) through a simple click of the "Split View" button. It also allows for side-by-side views for comparing gene expression patterns or tissue slice characteristics across multi-samples in ST experiments. To further enhance the comparative visualization capabilities of multiple features, we have developed the scMultiView component. It offers a wide range of visualization options, including violin plot, spatial feature plot, heatmap, dot plot, embedding plot, and motif logo plot (Figures 2F–2K).

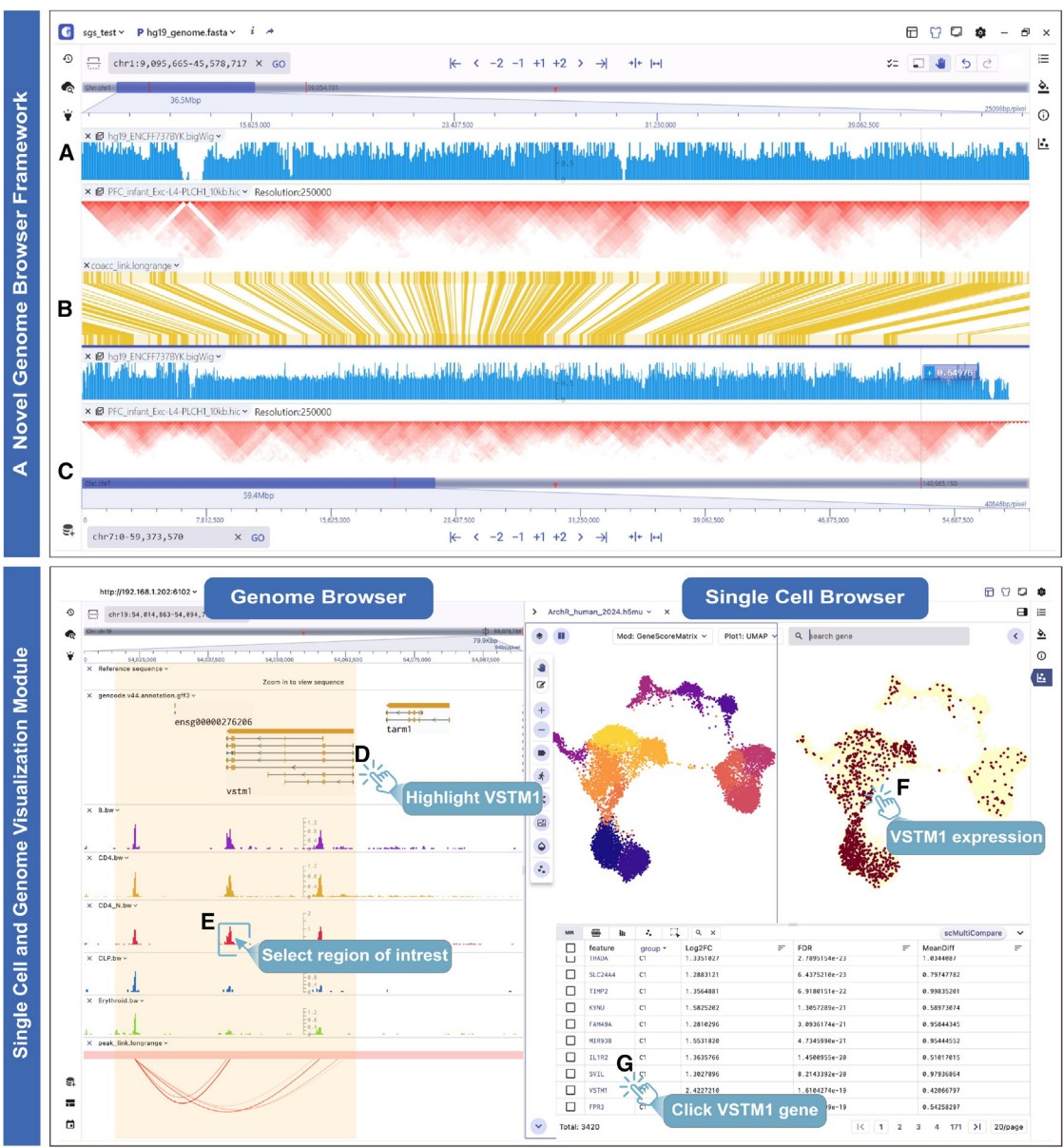

**Figure 3. Main features of SG mode**

(A–C) Double-chromosome visualization mode of SGS. (A) and (C) present bigwig and Hi-C interaction tracks on chr1 and chr7, respectively. The central bi-ginteract track (B) shows interaction-derived links between genomic regions.

(D and E) Visualization of human hematopoietic scATAC datasets in SG mode, featuring the genomic tracks for the *VSTM1* gene structure (D) and cell-specificity chromatin accessibility signals, along with gene CRE links (E).

(F) Cell embedding and *VSTM1* gene expression distribution in a single cell.

(G) Clicking on the *VSTM1* button in the marker table allows exploration of gene activity scores and genome-mapped signals.

See also Figures S10 and S11.

epigenetic signals (e.g., chromatin accessibility signals, differential peaks, gene *cis*-regulatory elements (CRE) links, methylation signals, chromatin interactions, sc-eQTL loci), and more (Figure S10). Additionally, it supports mainstream file formats and common operations and offers innovative features such as optimized display, dual-chromosome visualization, and efficient gene or region navigation. The dual-chromosome visualization mode enables robust comparative visual comparisons of epigenomic signals. By utilizing a double-chromosome display strategy, this mode provides several advantages. First, the top and bottom coordinates can cover different genomic regions, enabling the visualization of single-cell long-range interactions (Figures 3A–3C). Second, users can independently shift or zoom these coordinate regions, facilitating the comparative

visualization of cell-type-specific epigenomic signal differences in multiple regions simultaneously. Furthermore, users can freely arrange and customize these components according to their needs, enabling synchronized views of multimodal datasets or different views of the same data modality (Figures S11A and S11B). The integrative panel design of the SG visualization mode holds the potential for uncovering intricate relationships and interactions among diverse types of molecular information within distinct omics layers.

To gain a deeper understanding of the intermolecular dynamics between epigenomic modifications and cellular heterogeneity, SGS has enhanced the synergistic visualization of single-cell and genome browser panels. It utilizes genome features (e.g., peaks, genes, sc-eQTL) as interactive anchors to achieve coordinated visualization between these panels. When users click on these features in the marker table, the visualization panels automatically query, navigate, and render the dynamic changes of these features at both the genomic and single-cell levels. For example, the SG panel displays human hematopoietic datasets from previous studies.[31] By clicking on the *VSTM1* gene in the single-cell panel's marker table (Figure 3G), users can immediately navigate to the gene region to explore the distribution of epigenomic modification signals, such as chromatin accessibility and peak-to-gene links, in the genome browser (Figures 3D and 3E) and observe the expression patterns across different cell types (Figure 3F). In addition, for scATAC data, SGS allows users to select specific cell clusters in the single-cell panel and display the corresponding chromatin accessibility signals in the genome browser panel in real time. These interactive designs facilitate synchronous exploration of identical data types with linked views and provide a deep-insight view of complex multimodal data that is not possible by using disconnected tools side by side.

Overall, the SG visualization mode provides a scalable solution for integrating an ever-expanding volume of epigenomic multimodal datasets. By integrating the novel genome browser framework with single-cell panels, double-chromosome visualization, and multi-panel adaptive communication mechanisms, SG enables synchronous exploration and comparison of various datasets (e.g., snATAC, scM&T-seq, sn-m3C-seq, sc-eQTL) across linked panels.

### User case
#### sn-m3C-seq data visualization
The first example shows how SGS enables the synchronized visualization of sn-m3C-seq data encompassing DNA methylation and 3D chromatin conformation during the development of the human PFC and HPC.[32] In addition, we compared the visualization of this dataset using existing integrated visualization tools like Vitessce to further highlight SGS's unique value in visualizing sn-m3C-seq epigenomic multimodal data (Figure S2). Figure 4A displays two different views that integrate features selected in the SG visualization mode. (1) The genome browser displays the gene structure of the *RORB* gene (Figure 4A). Below this view, the panel shows the CG methylation signals alongside the Hic track of specific cell types (e.g., PFC 2T RG-1, PFC adult L1-3 NRXN2, PFC adult L4-5 FOXP2) (Figure 4B). (2) The single-cell panel presents the cell atlas from 13

developmental adult PFC and 9 HPC sn-m3C-seq samples (Figure 4C), showing the distribution of 10 primary cell types and *RORB* gene CG methylation patterns. Clicking on the *RORB* gene within the marker gene table, users can navigate to the corresponding region in the genome browser. This allows for the observation of the noticeably enhanced chromatin interaction strength specifically in the adult PFC L4-5 FOXP2 cell population within the *RORB* region, accompanied by a decreased CG methylation signal compared to other cell populations. Alternatively, in the single-cell panel, users can investigate the heterogeneity of the CG methylation signal distribution among various cell populations. Notably, a decrease is observed especially in excitatory neurons within the PFC L4-5 FOXP2 cell cluster, which is consistent with previous research findings (Figure 4D). This case demonstrates the powerful capabilities of SGS in visualizing complex epigenomic multimodal data. Through multi-panel integration and adaptive communication, SGS enables the integrated visualization of epigenome genome-mapped signals (gene annotation, DNA methylation, 3D chromatin conformation) and single-cell methylation information within a unified interface.

#### Visualization of 10x Genomics Visium multi-sample data
The second example illustrates the comparative visualization of 10x Genomics Visium multi-sample data from the anterior and posterior regions of the mouse brain using SGS. The scCompare function of SGS allows side-by-side viewing of distinct tissue slices (Figures 4E and 4F), revealing the spatial context and distribution of cells, such as the presence of cell cluster 1 specifically in the anterior brain (Figure 4G). Users can also utilize this function to observe the expression patterns of target genes across different tissue slice regions. For example, our observations reveal that genes such as *Stx1a, Prkcd*, *Hpca*, and *Ttr* exhibit specific expression in the cortex, thalamus, HPC, and choroid plexus regions, highlighting the spatial heterogeneity of gene expression within tissues (Figure 4H). The bottom panel of the SC mode shows thumbnails of multiple sample slices. Users can select any sample slice from the bottom panel and quickly identify the differences in cell proportions by clicking the meta-chart button to generate a stacked bar chart. For example, using the sample ID as the grouping feature, we obtain the proportion differences of cell clusters across anterior and posterior brain tissues. Compared to the posterior brain sample, we observe that the anterior brain sample has a lower proportion of cells in cluster 0 and a higher proportion of cells in cluster 2 (Figure S12).

The scMultiView component enables the batch comparison visualization of multiple genes, such as *Hpca*, *Prkcd*, *Stx1a*, *Ttr*, and *Penk*, through various charts like scatterplots, violin plots, dot plots, or heatmaps (Figure S13). This facilitates cell-type annotation and in-depth exploration of heterogeneity in gene expression.

#### sc-eQTL data visualization
The SGS provides researchers with a powerful tool for exploring sc-eQTL data, providing valuable insights into the impact of gene regulation on cellular function and disease mechanisms. It presents sc-eQTL data from the OneK1K research, including 982 individuals.[33] Additionally, we compared the functionality of SGS with the current mainstream tool, Vitessce, using the same dataset to highlight SGS's strengths in visualizing multimodal sc-eQTL data (Figure S3). The genome browser displays

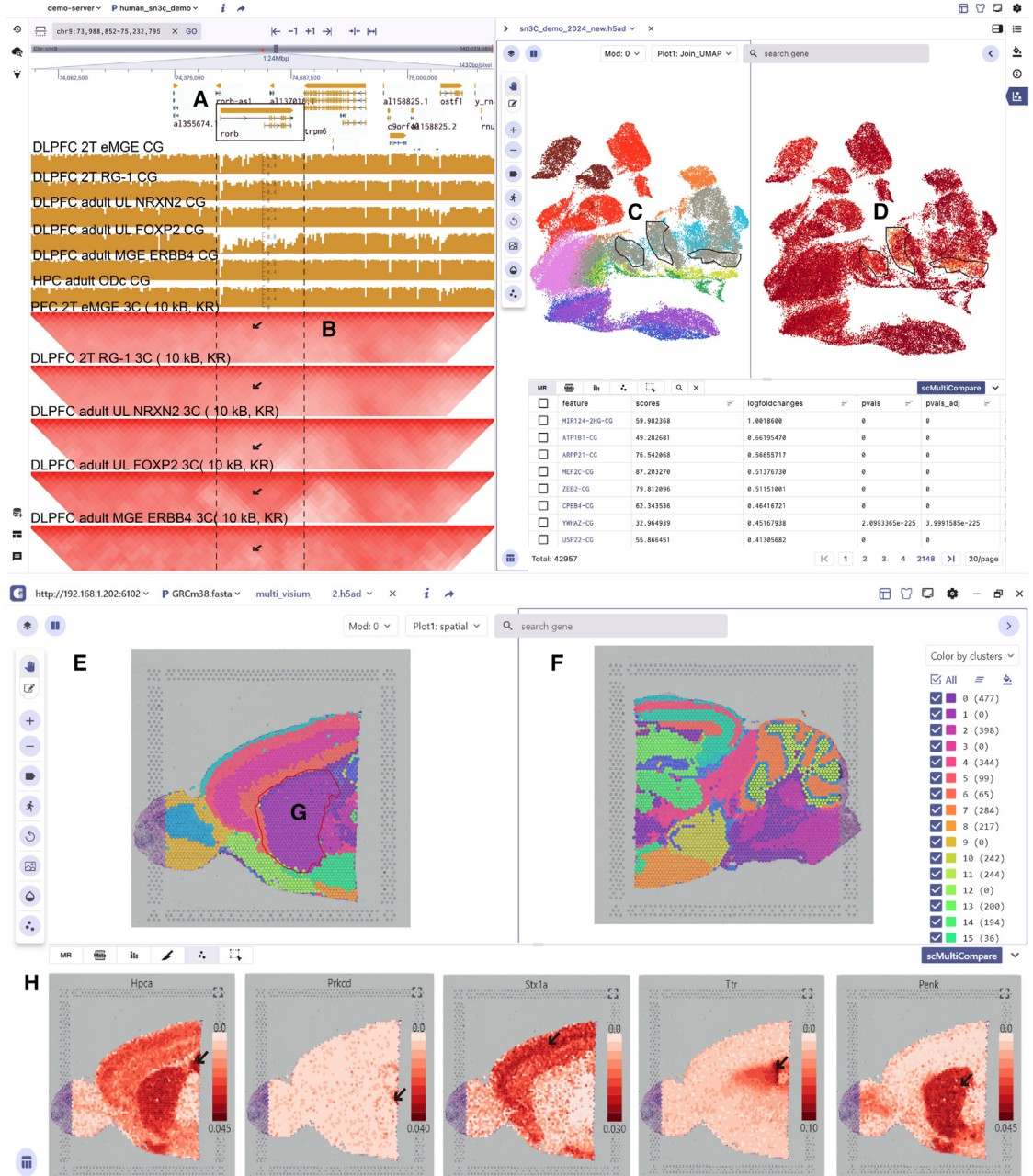

**Figure 4. Visualization of sn-m3C-seq data and 10x Genomics Visium multi-sample data in SGS**

(A and B) The sn-m3C-seq data visualization in SGS, with the left genome browser that displays the *RORB* gene annotation (A), HiC, and CG methylation signals for cell types (B).

(C) The cell atlas from PFC and HPC.

(D) The CG methylation patterns of the *RORB* gene.

(E and F) Split-screen visualization of 10x Genomics Visium data from mouse anterior (E) and posterior (F) brain tissues in SC mode.

(G) Cluster 1 is present mainly in the anterior brain tissue.

(H) Spatial feature plots of marker genes, including *HPCA* (hippocampal), *Prkcd* (thalamic), *STX1A* (cortex), and *Ttr* (choroid plexus), are presented at the bottom. See also Figures S2, S12 and S13.

multiple GWAS tracks of sc-eQTLs across various cell types. Users can easily locate specific sc-eQTL regions by entering gene symbols (e.g., *BLK*) or genomic positions (e.g.,

chr8:11300703-11611527) (Figures 5A and 5B). The single-cell visualization component displays the UMAP of 14 major cell types, while the interactive table at the bottom lists sc-eGenes

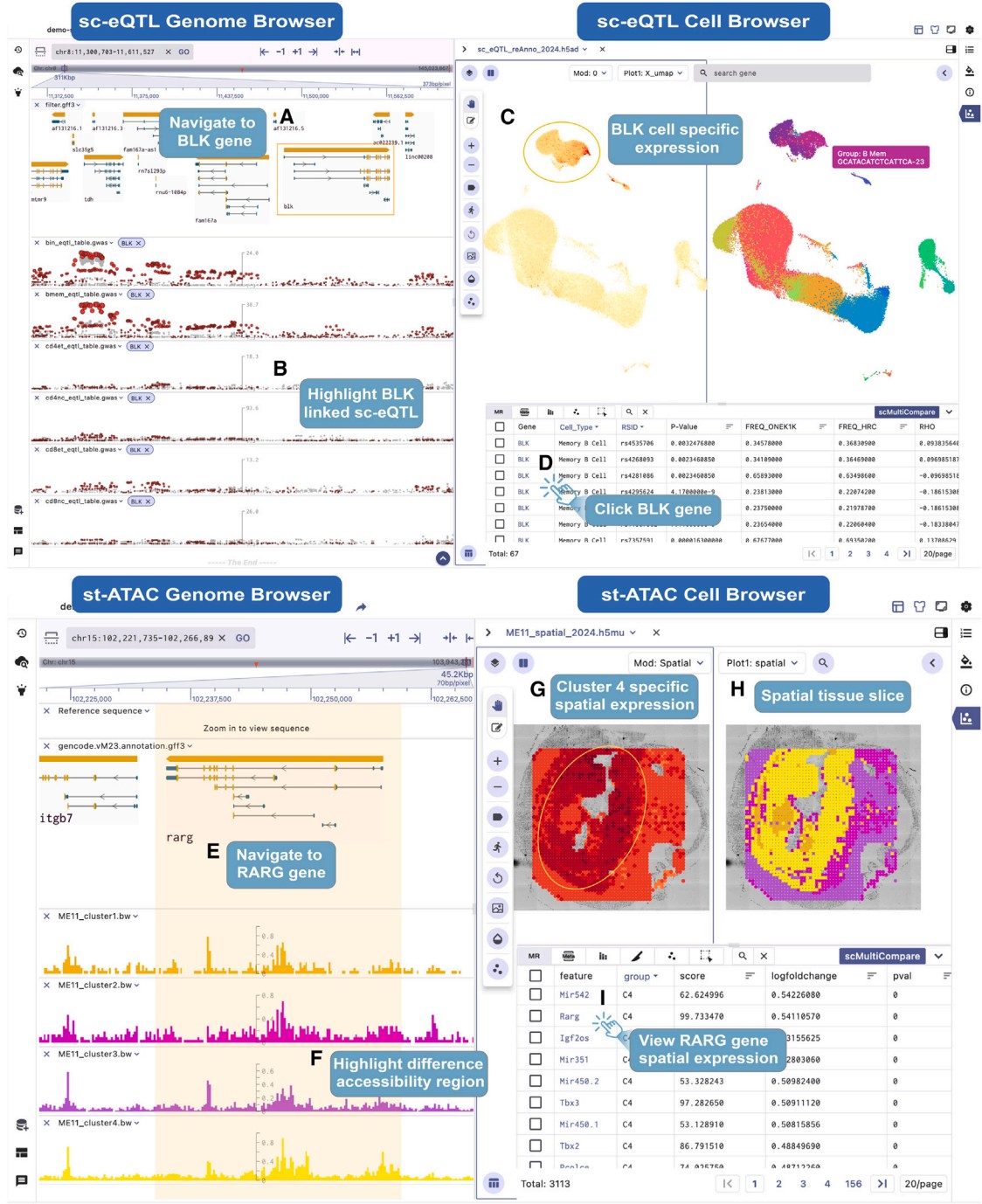

**Figure 5. Visualization of sc-eQTL data and spatial ATAC-seq data in SGS**

(A) Screenshot of the OneK1K sc-eQTL visualization demonstration. The genome framework shows the structure of the *BLK* gene.

(B) Highlighting cell-type-specific sc-eQTL loci. The significance of each locus is indicated by point size and height, with *BLK* gene-associated sc-eQTLs highlighted by red circles.

(C) The UMAP of 14 cell clusters and *BLK* gene expression pattern across different cell populations.

(D) Clicking on the *BLK* gene in the marker table to view synchronized changes in the single-cell and genome browser views.

(E) Integrative visualization of spatial ATAC-seq data from 11-day mouse embryos in SGS, with the genome browser navigated to the *RARG* gene.

(F) The genome view shows the *RARG* gene and chromatin accessibility signals across various cell clusters, with cluster 4 showing the highest signal.

*(legend continued on next page)*

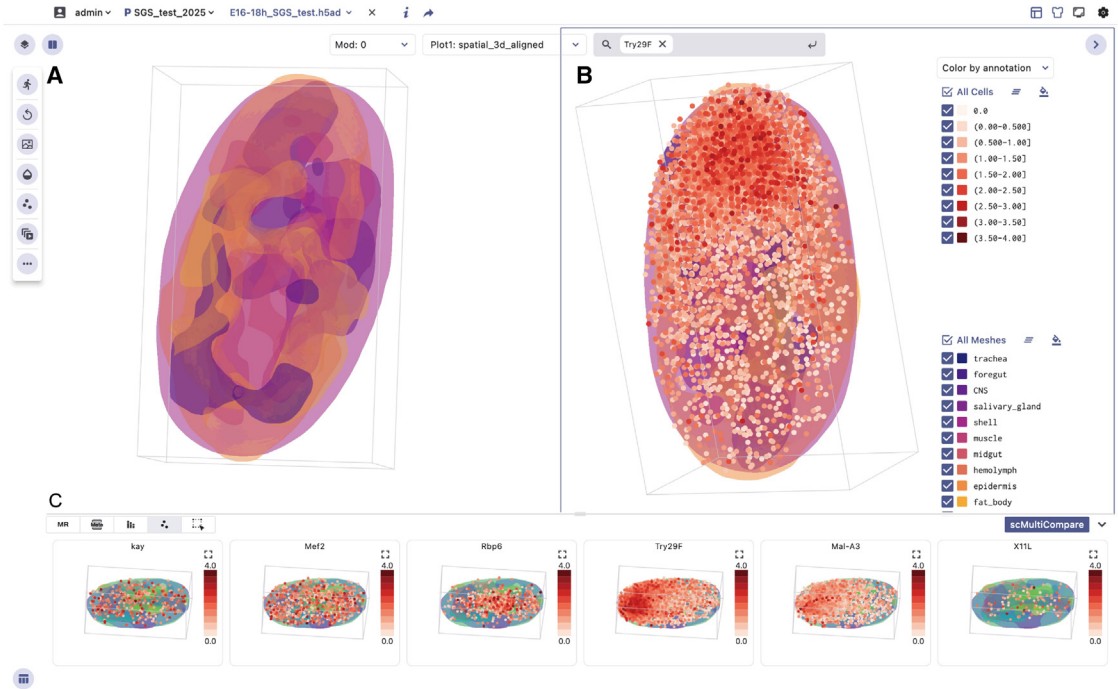

**Figure 6. Visualization of late-stage *Drosophila* embryo 3D SRT data via the SGS browser**
(A) The surface model of the *Drosophila* embryo tissue.
(B) The spatial expression distribution of *Try29F*.
(C) Spatial feature maps of marker genes, including *Mal-A3*, *Try29F*, and *Rbp6*.

(genes significantly associated with sc-eQTLs) related to specific cell types. Clicking on the *BLK* gene in the marker table allows users to explore its expression heterogeneity across cell types and navigate to the associated sc-eQTL region in the genome browser (Figure 5D). The single-cell panel reveals the high expression of this gene specifically in Bmem cells (Figure 5C). The genome browser panel highlights *BLK*-specific sc-eQTLs in red, with other sc-eQTLs in gray. Users can access details such as *BLK* sc-eQTL site ID, *p* value, and more. Here, we observe that many expressed SNPs associated with the *BLK* gene show correlations with *BLK* expression in CD4 NC, CD8 ET, CD8 NC, B Mem, and B IN cells. Notably, among the associated eQTL loci, the *rs2736336* locus was identified. Previous studies have reported that the *rs2736336* variant leads to differential expression of *BLK* in B Mem cells, indicating its potential role in inter-individual variability of B lymphocyte tolerance.

### Spatial-ATAC-seq data visualization

The SGS browser offers an integrated visualization of spatial-ATAC-seq data, enabling a comprehensive exploration of the spatial chromatin accessibility landscape during mouse embryo development (ME11).[34] Users can select and group chromatin accessibility signal tracks. By clicking the navigation button for a target gene, users can quickly locate the genomic region and

view accessibility signals across cell clusters. For example, by clicking the rapid navigation button for the *RARG* gene (Figure 5I), users can navigate to the genomic region and see that the *RARG* gene exhibits higher chromatin accessibility signals in cluster 4 (Figures 5E and 5F). Among these clusters, the *RARG* gene (associated with prominent facial features and limb development in the embryo) is extensively activated in cluster 4 (Figure 5G), indicating a potentially crucial role of the cluster in limb bud development, skeletal growth, and matrix balance. Coupled with tissue slice information, we identified four clusters with distinct spatial patterns in the spatial-ATAC-seq data of 11-day-old mouse embryos (Figure 5H).

This approach allows users to quickly observe epigenomic signals and spatial expression differences of target genes at the single-cell level in mouse embryos. Linking epigenetic signals to spatial context enhances the understanding of gene regulation and expression patterns.

### 3D SRT data visualization

To demonstrate the potential of the SGS browser in visualizing 3D SRT datasets, we utilized the scCompare functions to present the late-stage embryos spatial data of *Drosophila*.[35] In Figure 6A, we observed different regions of the *Drosophila* embryo, such as the central nervous system (CNS), midgut, fat

(G) The spatial view indicates that the *RARG* is highly expressed in cluster 4.
(H) The spatial view reveals the heterogeneity of cell composition.
(I) Clicking on the *RARG* gene in the marker table to view synchronized changes in both single-cell and genome browser views.
See also Figure S3.

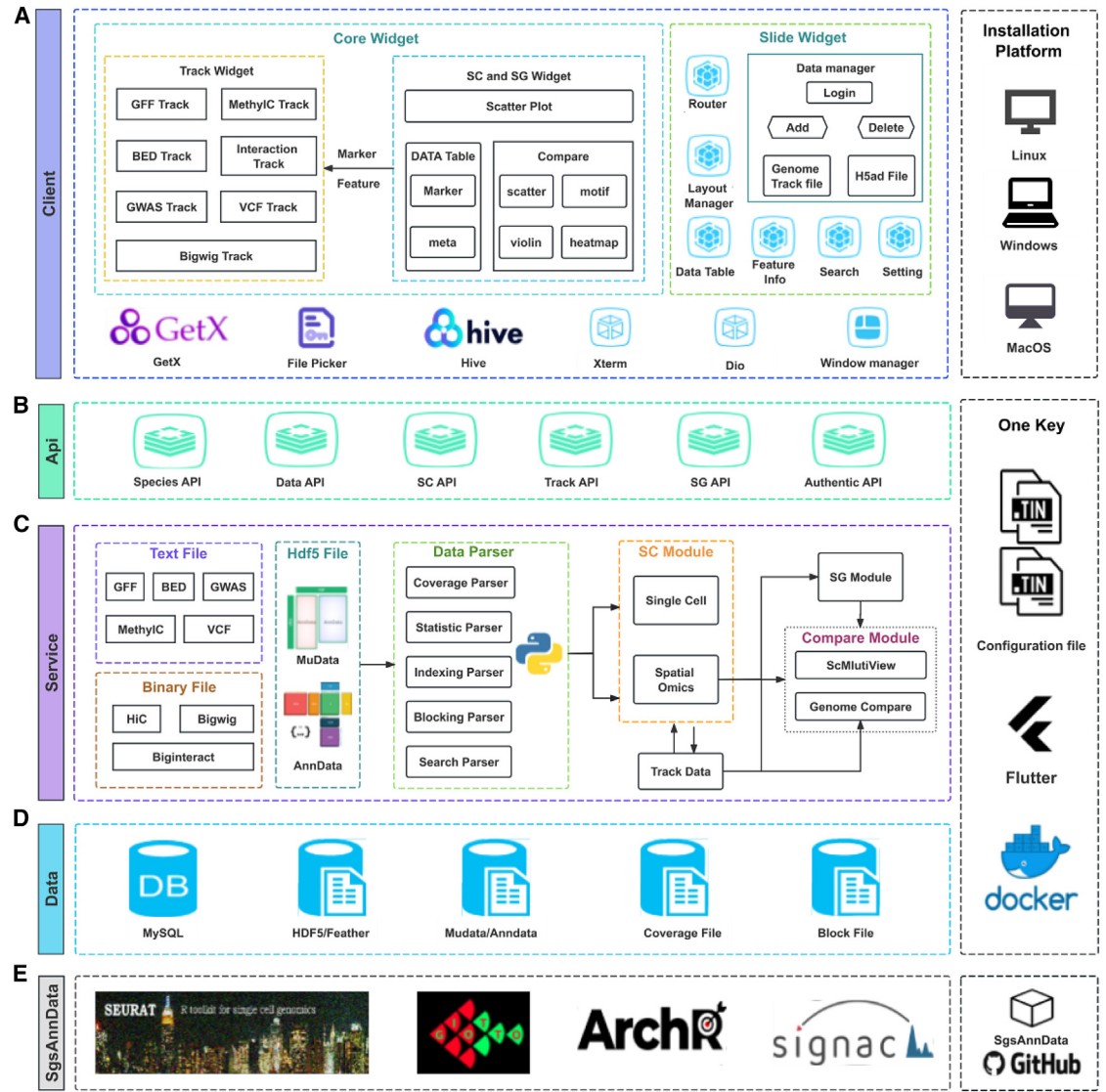

**Figure 7. The framework of SGS**
(A) Client built with Flutter, featuring Core and Slide Widgets, supporting Linux, Windows, and MacOS.
(B) The API layer is used for front-end and back-end communication.
(C) The back-end architecture, includes the Data Parser, SC Mode, SG Mode, and Comparison Mode.
(D) The data layer defines storage strategies for different types of data.
(E) SgsAnnData is an R package used for data format conversion.

body, and epidermis, by displaying surface models of the tissues. The surface models of different tissues are observed to be analogous to the anatomical structures of embryos and larvae (Figure 6A). In addition, SGS supports functionalities such as zooming, rotating, toggling between groups, and adjusting the size and transparency of cell points for 3D SRT datasets. These features provide users with an interactive visualization toolkit to explore 3D data along different body axes or from various perspectives of complex organs. Additionally, in Figure 6B, users can capture changes in spatial expression patterns within the endoderm, mesoderm, and ectoderm regions by searching for marker genes specific to each tissue. For example, searching

for *Mal-A3* reveals its predominant enrichment in the posterior midgut of the embryo; *Try29F* exhibits specific expression in the posterior midgut, indicating the dynamic regional expression patterns within gene families and suggesting functional regionalization dynamics in the midgut during embryogenesis (Figure 6B). Furthermore, previous studies have indicated that certain regulators are tissue specific and may play a role in tissue development, such as *Rbp6* in the CNS, *grh* in the epidermis, *srp* in the fat body, *kay* in the midgut, and *Mef2* in muscles (Figure 6C). Users can visualize multiple genes of interest simultaneously and compare expression pattern differences across different cell groupings using SGS's Compare function.

## DISCUSSION

### Overview

Comprehensive visualization of single-cell and spatial multiomics data is crucial for investigating the intermolecular dynamics between epigenomic regulation and transcriptomic or proteomic expression within individual cells across development, aging, and disease. This has led to the generation of large-scale, high-dimensional, and complex multimodal data, posing challenges for the visualization of scMulti-omics data. In this paper, we introduce SGS, which provides a unified interface for the joint visualization of scMulti-omics data, including genomics, transcriptomics, proteomics, and single-cell or spatial resolution epigenomics. It offers modular panels for heterogeneous data types, combined with multiscale interactive navigation and collaborative exploration. SGS has many advantages and novel features (Figure 7). First, its graphical installation and collaborative visualization capabilities reduce the learning curve, facilitating rapid installation and collaborative data visualization tasks like co-annotation, comments, and project management. Second, SGS supports interactive visualization of single-cell clustering, spatial tissue slices, and 3D data browsing with surface model plots. By leveraging the scCompare and scMultiView functions, SGS offers rich plotting capabilities, fostering comprehensive comparisons across modalities and spatial tissues, as well as multiple genes. The 3D data browser allows for interactive exploration of heterogeneous gene expression in 3D space, enhancing the depth of analysis and understanding of spatial gene expression patterns. Furthermore, to meet the demands of visualizing complex epigenomic multimodal data, SGS includes a novel genome framework. This framework accommodates various data formats, provides multiple visualization features, and incorporates dual-chromosome visualization capabilities to enhance the comparative visualization of genome-mapped data. By further integrating the single-cell component and genome browser framework within the unified panel, SGS enables integrated visualization of diverse data types. The synchronized response design between these panels holds significant value by enabling integrated visualization, cross-modal insights, interactive exploration, and enhanced data interpretation.

### Limitations of the study

With the upcoming massive wave of new and diverse scMulti-omics datasets, the visualization of different modalities within a unified interface and the deeper integration of data features from multiple modalities often require advanced integration algorithms. SGS currently offers an appealing solution through its multi-panel integration and adaptive communications. To enhance the exploration of high-dimensional data modalities, SGS will employ a comprehensive suite of integration strategies to effectively tackle the challenges in multimodal data integration. These challenges include batch effects, cross-platform normalization, data modality heterogeneity, and differences in resolution between spatial and single-cell data.[36,37] These strategies include vertical integration methods (e.g., MOJITOO,[38] scAI[39]), horizontal integration methods (e.g., totalVI,[40] UINMF[41]), and mosaic integration algorithms (e.g., totalVI, UINMF, scArches[42]), each tailored to address specific challenges.

In addition, as spatial sequencing technologies continue to advance, the visualization of data requires the extraction of signals from numerous image layers, each representing different fluorescent channels collected during various staining rounds. To enhance the ability to view individual molecular labels and effectively handle multi-layered image data, our efforts will focus on optimizing the rendering of imaging-based spatial transcriptomics[43] data using Open Microscopy Environment next-generation file formats.[44]

### Conclusions

In summary, SGS empowers researchers with a robust omics browser, unlocking novel insights from scMulti-omics data. With its exceptional flexibility, scalability, and accessibility, SGS plays a crucial role in advancing our understanding of cell-type differentiation, the underlying gene regulatory networks, and spatial heterogeneity. The platform holds promise for future advancements, accommodating new data modalities and integrating diverse omics datasets, thereby further propelling scientific discoveries in the field.

## RESOURCE AVAILABILITY

### Lead contact

Further information and requests for resources and code should be directed to and will be fulfilled by the corresponding authors Fang Lu (lufang0411@sina.com) and Yi Wang (yiwang28@swu.edu.cn).

### Materials availability

This study did not generate new unique reagents.

### Data and code availability

The mouse embryo data from Chen et al.[45] can be accessed at https://db.cngb.org/stomics/mosta/. The human lung spatial multi-slice data from He et al.[46] can be accessed at https://genome.ucsc.edu/s/brianpenghe/scATAC_fetal_lung20211206. The human hematopoietic stem cell scATAC data can be accessed at https://www.archrproject.com/. The human PFC and HPC sn-m3C-seq data from Heffel et al.[32] can be accessed at https://cells.ucsc.edu/?ds=brain-epigenome+human-brain-m3c. The mouse brain spatial multi-sample data can be accessed at https://www.10xgenomics.com/datasets?menu%5Bproducts.name%5D=Spatial%20Gene%20Expression. The human OneK1K sc-eQTL data from Yazar et al.[33] can be accessed at https://cellxgene.cziscience.com/collections/dde06e0f-ab3b-46be-96a2-a8082383c4a1. The eQTL data were downloaded from https://onek1k.s3.ap-southeast-2.amazonaws.com/onek1k_eqtl_dataset.zip. ME11 spatial-ATAC-seq data from Deng et al.[34] were accessed via GEO: GSE171943. The PBMCs 5K scATAC-seq data were downloaded from https://cf.10xgenomics.com/samples/cell-atac/2.0.0/atac_pbmc_5k_nextgem/. The Drosophila embryos 3D SRT data from Guo et al.[47] can be accessed at https://www.bgiocean.com/vt3d_example/. The human heart 3D SRT data from Asp et al.[48] can be accessed at https://www.bgiocean.com/vt3d_example/. The next-generation sequencing demonstration data can be accessed at https://epigenomegateway.wustl.edu/. The code for SGS is available at GitHub (https://github.com/fanglu0411/sgs) or Zenodo (https://doi.org/10.5281/zenodo.14963606). Tutorials are available at https://sgs.bioinfotoolkits.net/home.

## ACKNOWLEDGMENTS

We thank Zhaoyuan Wei and Guoqing Zhang at Southwest University for critical reading of the manuscript. The study was funded by the National Natural Science Foundation of China (grant nos. U21A20248 and 32000340) and the

Fundamental Research Funds for the Central Universities (grant no. XDJK2019TJ003).

## AUTHOR CONTRIBUTIONS

T.X. and Y.W. conceptualized the project, and Y.W. supervised the project. T.X., J.S., F.L., and Y.L. designed and wrote the software code. T.X., J.S., and Y.M. conducted the software testing and analysis. T.X. and J.S. performed the data collection and labeling and wrote the original draft of the paper. L.X. reviewed the manuscript. Y.W. finalized the manuscript, with input from all authors; all the authors read and approved the final manuscript.

## DECLARATION OF INTERESTS

The authors declare no competing interests.

## STAR★METHODS

Detailed methods are provided in the online version of this paper and include the following:

- KEY RESOURCES TABLE
- METHOD DETAILS
  - Software architecture and implementation
  - Client design
  - Server design
  - Single-cell and spatial multi-omics visualization component
  - SgsAnnData

## SUPPLEMENTAL INFORMATION

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

**Cell Genomics**
Technology

# STAR★METHODS

## KEY RESOURCES TABLE

| REAGENT or RESOURCE | SOURCE | IDENTIFIER |
|---|---|---|
| **Deposited data** | | |
| Mouse embryo Stereo-seq data | Chen et al.[45] | https://db.cngb.org/stomics/mosta/ |
| Human lung spatial multi-slice data | He et al.[46] | https://genome.ucsc.edu/s/brianpenghe/scATAC_fetal_lung20211206 |
| Human hematopoietic stem cell scATAC data | Granja et al.[31] | https://www.ncbi.nlm.nih.gov/geo/query/acc.cgi?acc=GSE139369 |
| Human sn-m3C-seq data | Heffel et al.[32] | https://cells.ucsc.edu/?ds=brain-epigenome+human-brain-m3c |
| Mouse brain 10X Visium data | 10x genomics | https://www.10xgenomics.com/datasets?menu%5Bproducts.name%5D=Spatial%20Gene%20Expression |
| Human oneK1K sc-eQTL data | Yazar et al.[33] | https://pubmed.ncbi.nlm.nih.gov/35389779/ |
| ME11 Spatial-ATAC-seq data | Deng et al.[34] | https://www.ncbi.nlm.nih.gov/geo/query/acc.cgi?acc=GSE171943 |
| PBMCs 5K scATAC-seq data | 10x genomics | https://cf.10xgenomics.com/samples/cell-atac/2.0.0/atac_pbmc_5k_nextgem/ |
| Drosophila embryos 3D SRT data | Guo et al.[47] | https://www.bgiocean.com/vt3d_example/ |
| NGS demo data | WashU Epigenome Browser | https://epigenomegateway.wustl.edu/ |
| Human heart 3D SRT data | Asp et al.[48] | https://www.bgiocean.com/vt3d_example/ |
| **Software and algorithms** | | |
| Flutter 3.16 | Flutter Framework | https://gaussian.com |
| dio 5.6.0 | Flutter Package | https://www.python.org |
| SGS | This paper | https://doi.org/10.5281/zenodo.14963606 |
| File Picker 8.1.2 | Flutter Package | https://www.perkinelmer.com/category/chemdraw |
| Flask 3.02 | Python Software | https://flask.palletsprojects.com/en/stable/ |
| MySQL 8.0 | Database Tool | https://www.mysql.com/cn/downloads/ |
| tabix 1.7–2 | Python Software | https://github.com/samtools/tabix |
| pybigwig 0.3.22 | Python Software | https://github.com/deeptools/pyBigWig |
| Straw 0.1.0 | Durand et al.[49] | https://github.com/aidenlab/straw |
| anndata 0.8.0 | Virshup et al.[50] | https://github.com/scverse/anndata |
| mudata 0.2.3 | Bredikhin et al.[51] | https://github.com/scverse/mudata |
| Getx 4.6.6 | Flutter Package | https://github.com/jonataslaw/getx |
| Seurat 5.1.0 | Satija et al.[27] | SCR_016341 https://satijalab.org/seurat/ |
| Signac 1.14.0 | Stuart et al.[29] | https://github.com/stuart-lab/signac |
| Giotto 4.0.5 | Del Rossi et al.[30] | https://github.com/drieslab/Giotto |
| ArchR 1.0.2 | Granja et al.[28] | https://github.com/GreenleafLab/ArchR |

## METHOD DETAILS

### Software architecture and implementation

The SGS browser adopts a front-end and back-end separation architecture. The software is divided into distinct layers, including the data layer, service layer, API layer, and client layer. The Flask-based service layer handles data parsing and processing. It efficiently handles data-related tasks within the SGS software. The client layer utilizes the Flutter framework to enable interactive visualization (Figure 7A). The API layer is mainly used for the communication between the service layer and the client layer (Figure 7B).

## Client design

The SGS browser client is built using the Flutter framework (v. 3.16), known for its robust cross-platform capabilities, and can operate seamlessly on MacOS, Windows, and Linux. We employ dio (v. 5.6.0) for HTTP service handling, File Picker (v. 8.1.2) for file uploads, and Xterm for interactive terminal interfaces. Leveraging Dart as the primary programming language, we utilize its rich functionality and extension library support to implement the features and interface of the SGS browser client. The SGS client comprises two major widgets: Core Widgets and Side Widgets. The Core Widgets consist of three parts: the Genome Browser Widget, the SG Widget, and the SC Widget. The Side Widgets primarily include the Data Table Widget, Search Widget, and Feature Info Widget, providing auxiliary functions for region highlighting, track list operations, theme, and layout settings.

## Server design

The back-end uses the Flask (v. 3.02) framework. The back-end server is primarily divided into four main components: file type identification and management, data parsing and calculation, data relationship mapping, and mode communication. Moreover, it utilizes a multi-threaded approach to facilitate concurrent data processing, enabling batch data uploads and enhancing data addition efficiency (Figure 7C). For efficient data management and storage, we have employed MySQL (v. 8.0) as our preferred solution. SGS leverages advanced techniques such as memory mapping and index building to optimize the performance of rapid queries, particularly in track transformation and the scaling of extensive datasets (Figure 7D).

## Single-cell and spatial multi-omics visualization component
### *A novel genome browser framework*

The genome browser framework supports popular data formats. In the back-end, we implemented specific parsing strategies for binary and text file formats. For text formats (e.g., BED, VCF and GFF), we utilized tabix (v. 1.7–2) to retrieve region-specific data. Binary formats (e.g., Bigwig and Hic) are parsed via pybigwig (v. 0.3.22) and straw[49] (v. 0.1.0). Data storage employs the feather memory mapping technique. In the front-end, we simplified visualization by implementing a universal Track Widget parent class. Customized SGS chart libraries support various display styles including bar charts, gene structures, area charts, and heatmaps. Techniques such as preloading view data, dynamic retrieval, memory management, and minimize storage consumption ensured rapid feature rendering and zooming at different visualization levels. To achieve rapid scaling and visualization rendering, we defined distinct zoom levels based on feature density. For each view resolution scale (termed the "zoom level"), we dynamically partitioned the data based on the current visual interval chromosome length (*CurrentRange*) and view width (*ViewWidth*) to compute the *realScale*.

$$realScale = \frac{ViewWidth}{CurrentRange}$$

To determine the current zoom level (*currentScale*), we identify the value from the predefined target scaling set (*desiredScales*) that minimizes the error between the chromosome length (*realScale*) corresponding to each pixel in the current view using the following formula:

$$currentScale = findScale(realScale, desiredScales)$$

Each data block is defined with a *BlockWidth* of 1000 pixels. Subsequently, we calculated the length of each block in base pairs (*blockBPs*) at the current zoom level using the following formula:

$$blockBPs = \frac{BlockWidth}{currentScale}$$

Calculate information for individual data blocks, start position and ending position:

$$BlockInfo_i = \{index_i, start\_bp_i, end\_bp_i\}$$

$$start\_bp_i = i * blockBPs$$

$$end\_bp_i = start\_bp_i + blockBPs$$

Where the *BlockInfo_i* represents the information of each data block, including block index and the starting and ending bases. *start_bp_i* represents the starting base of *block_i*. *end_bp_i* represents the ending base of the *block_i*, determined by adding the size of the block to the starting base.

### *Single-cell component*

The single-cell component includes the SC and SG Widgets. We use AnnData[50] and MuData[51] for data storage, and the Zarr format for large-scale multimodal data. Furthermore, to optimize marker feature queries, we utilize the N-gram language model (https://github.com/joshualoehr/ngram-language-model), which is a language model that utilizes n-grams to capture the contextual dependencies between words in a given dataset. To optimize the rendering speed and minimize memory consumption, we employ methods such as cell grouping rendering, gradient interval division, and random sampling. We defined a maximum rendering cell

number (*MaxDrawingCells*). When the number of cells exceeds this threshold, we perform proportional random downsampling of each cell group via the following formula:

$$N'' = \begin{cases} N & if\ N \leq MaxDrawingCells \\ MaxDrawingCells & if\ N > MaxDrawingCells \end{cases}$$

where the $N$ represents the actual number of cells, $N''$ represents the number of cells after subsampling.

For spatial transcriptomic data, the precise alignment of spatial coordinates with tissue slices is ensured by using scaling factors for different image resolutions. Additionally, a caching mechanism preloads spatial slice information, enabling swift loading and efficient visualization for comparing multiple samples. The scMultiView Widget supports various visualization plots, including violin plot, boxplot, heatmap, and dot plot.

### *Multi-panel adaptive communication*

We developed a multi-panel communication mechanism that integrates and coordinates single-cell, spatial, and genome visualization panels, enabling data exchange, collaborative exploration, rapid navigation, and highlighting across variant components. Using Getx (v. 4.6.6) for inter-panel communication state and route management, we established an adaptive communication system between the single-cell and the genome browser panels. The system consists of a unified communication protocol, a core communication engine, and functional mode components. Each functional mode component includes communication triggers and listeners, while the unified communication protocol defines data transmission methods and network connection rules.

The core communication engine coordinates signals between the genome browser and single-cell components. It manages data transmission, error handling, connection management, and security authentication. The communication triggers enable user interactions, such as clicks in the single-cell panel to select marker features (such as genes, peaks, and sc-eQTLs). These actions are then transmitted to the core engine, which subsequently directs them to the communication listener within the genome browser. Upon receiving the signal, the listener swiftly retrieves and locates the position of the marker feature, synchronously highlighting the region and displaying other genomic signals such as methylation signals, and chromatin interaction intensities, etc.

### SgsAnnData

We developed the SgsAnnData R package to convert outputs from mainstream software (e.g., Seurat, ArchR) into AnnData format for compatibility (Figure 7E). The tutorial can be accessed at https://github.com/bio-xtt/SgsAnnDataV2/tree/main.

