## [Document S2. Transparent peer review records for Xia et al. · Cell Genomics]

Empowering Integrative and Collaborative Exploration of Single-Cell and Spatial Multimodal Data with SGS Genome Browser

Tingting Xia, Jiahe Sun, Yongjiang Luo, Hailong Guo, Yudi Mao, Ling Xu, Fang Lu, and Yi Wang

Summary

Initial submission: Received : Sep 23, 2024

Scientific editor: Sara Rohban

First round of review: Number of reviewers: 3
Revision invited : Oct 30, 2024
Revision received : Jan 15, 2025

Second round of review: Number of reviewers: 2
Revision invited : Feb 19, 2025
Revision received : Feb 23, 2025

Third round of review: Number of reviewers: 1
Accepted : Mar 17, 2025

Data freely available: YES

Code freely available: YES

This transparent peer review record is not systematically proofread, type-set, or edited. Special characters, formatting, and equations may fail to render properly. Standard procedural text within the editor's letters has been deleted for the sake of brevity, but all official correspondence specific to the manuscript has been preserved.

Referees' reports, first round of review

Reviewer #1:

This manuscript presents the SGS Genome Browser for visualizing single-cell and spatial transcriptomics data. I have visited the SGS website, watched the tutorial videos, downloaded the SGS software to my local computer, and tried some example analyses. The software appears to be well-designed and easy to use. The ability to run SGS both via a web page and locally is extremely useful. The videos are also informative, and it is evident that the authors have invested considerable time and effort in developing the software.

That being said, while this tool could be valuable for addressing many pressing needs in data visualization and exploration, its functionality unfortunately overlaps significantly with Vitessce, a recently published visualization tool (<https://www.nature.com/articles/s41592-024-02436-x>). Below are my specific comments:

1. The SGS Genome Browser appears to be similar to Vitessce, which has already been published. In the introduction, the authors state that "it is important to acknowledge that these tools often require users with programming skills, making them less accessible to non-programmers. Furthermore, both tools have notable constraints in effectively visualizing epigenomic multimodal data." This description is too vague, and the differences between SGS and Vitessce are not mentioned anywhere in the manuscript, except for Supplementary Table 1. The authors should clearly detail the advantages of the SGS Genome Browser over Vitessce, particularly in the real data example.
2. In Supplementary Table 1, the only differences between SGS and Vitessce seem to be the session URL sharing function and the accepted data formats. These differences alone may not be sufficient to justify the publication of SGS in a top journal.
3. Based on the description in the manuscript, the association between the SC and SG modules appears weak. It would be much more useful if users could select certain cells in the SC module and display the epigenomic information for those selected cells in the SG module. Otherwise, users could simply open two visualizers side-by-side (e.g., Loupe Browser for single-cell data and UCSC Genome Browser for epigenomic data).
4. For single-cell multi-omics data that profile both gene expression and chromatin accessibility in the same cells, can SGS display gene-CRE linkages?
5. Can SGS input data already processed by other software, such as Seurat and Scanpy?
6. Without watching the tutorial videos, it is quite difficult to begin using SGS. For instance, it seems that users must first deploy SGS before loading data and performing analyses or visualizations. This process seems tedious, and some users may feel intimidated and stop at this point. I wonder if there is a way to simplify the process of initializing an analysis?

Reviewer #2:

The paper presents the SGS Genome Browser, a novel tool designed to facilitate the integrative and collaborative exploration of single-cell and spatial multimodal data. With the growing complexity of data generated from single-cell and spatial technologies, there is a need for tools that can help researchers visualize, analyze, and compare features from multiple modalities (e.g., RNA expression, protein levels, DNA accessibility, etc).

However, there are parts that could be clarified to strengthen the manuscript:

1. While collaboration is mentioned as a core feature, there are limited specific details or examples showing how real-time collaboration works, especially in terms of data security, version control, or simultaneous editing.

It will be helpful to provide demonstrations of how collaboration is implemented (e.g., how users can annotate the same dataset in real-time, how conflicts are resolved, etc.), along with information on security and user permissions in collaborative settings.

2. The paper may underplay the challenges of multimodal data integration, such as dealing with batch effects, cross-platform normalization, or differences in resolution between spatial and single-cell data. It will be helpful to address these challenges more explicitly, discussing how SGS deals with such issues, or include future directions for tackling these common obstacles in multimodal data analysis.

3. In the conclusion section, authors mentioned: "SGS plays a crucial role in advancing our understanding of differentiation trajectories, the underlying gene regulatory networks, cell-to-cell interactions, microenvironmental spatial organization, cellular lineages, and clonal dynamics". It is not clear whether trajectory analysis, regulatory network, or cell-to-cell interactions are supported by SGS, in terms of visualization.

Other minor modification suggestions are listed below:

In Abstract/Summary, multimodal -> multimodal

Figure 1C, sCompare -> scCompare

Page 5, Supplementary Figure 8 -> Supplementary Figure 9

Page 18, The back-end using Flask (version 3.02) framework, is famous for its lightweight -> The back-end uses Flask (version 3.02) framework, which is famous for its lightweight

Page 22, To compatible with the output of these tools -> To be compatible with the output of these tools

Reviewer #3:

Summarize :

The authors developed a new browser called SGS, which can visualize large-scale, high-dimensional and complex data in single-cell and spatial omics, and provides functions such as multi-panel to visualize and interact with multimodal data. It is valuable for researchers as it simplifies the visualization of complex data, making it accessible to those with limited programming skills.

Minor issues:

1. Paragraph 3 in Introduction section: Authors mentioned that "SGS supports various data formats including AnnData/AnnData.zarr, MuData/MuData.zarr". The software is very friendly to researchers without programming skills. However, compared with the data formats supported by genomic data, the data formats supported by single-cell and spatial group data are relatively few. For example, data downloaded from public databases may contain data in other formats such as rds, gef, and h5, which cannot be easily loaded using SGS software.
2. Paragraph 1 in Results section: Authors mentioned that "This makes SGS compatible with multiple systems, including Linux, Windows, MacOS, and Android." In the download link (<https://sgs.bioinfotoolkits.net/home>), the Android platform tool is not available, and Android platform is not mentioned again in the article.

Authors' response to the first round of review

RESPONSE TO REVIEWER COMMENTS:

We thank the Reviewers for the time and effort spent carefully reviewing our manuscript and providing constructive comments. Point-by-point responses to all comments and modifications to the manuscript are listed below. The reviewers' comments are in plain text, and our responses are in blue. The cross-references to the manuscript are bold and underlined. (Line numbers mentioned in the responses may not coincide with the original line numbers.)

Reviewer #1:

This manuscript presents the SGS Genome Browser for visualizing single-cell and spatial transcriptomics data. I have visited the SGS website, watched the tutorial videos, downloaded the SGS software to my local computer, and tried some example analyses. The software appears to be well-designed and easy to use. The ability to run SGS both via a web page and locally is extremely useful. The videos are also informative, and it is evident that the authors have invested considerable time and effort in developing the software.

That being said, while this tool could be valuable for addressing many pressing needs in data visualization and exploration, its functionality unfortunately overlaps significantly with Vitessce, a recently published visualization tool (<https://www.nature.com/articles/s41592-024-02436-x>). Below are my specific comments:

1. The SGS Genome Browser appears to be similar to Vitesce, which has already been published. In the introduction, the authors state that "it is important to acknowledge that these tools often require users with programming skills, making them less accessible to non-programmers. Furthermore, both tools have notable constraints in effectively visualizing epigenomic multimodal data." This description is too vague, and the differences between SGS and Vitesce are not mentioned anywhere in the manuscript, except for Supplementary Table 1. The authors should clearly detail the advantages of the SGS Genome Browser over Vitesce, particularly in the real data example.

Response: Thank you for your constructive feedback on our manuscript. We appreciate your observation that the initial submission lacked a sufficient comparison between the SGS browser and Vitesce in our manuscript. Your feedback is very valuable in helping us improve and enhance our manuscript. We have adopted your suggestions and have made substantial revisions to address this concern. Please see our response below:

1. Advantages of SGS over Vitesce (1) Enhanced Visualization of Epigenomic Multimodal Data

Vitesce is primarily focused on the exploration of spatial single cell experiments data, SGS places greater emphasis on single-cell and spatial epigenomic multimodal data visualization. Its capabilities extend beyond single-cell and spatial omics data, making it a versatile standalone genome multi-omics visualization platform. In terms of genome-mapped data visualization, Vitesce utilizes the HiGlass framework for the visualization of gene structures and chromatin accessibility signals. However, its ability to display genome-mapped features is limited, with a primary focus on snapATAC analysis outcome visualization. It lacks precise navigation, querying capabilities, and detailed track feature information visualization that are crucial for in-depth analysis of single-cell and spatial epigenomic data. To address these limitations, we have developed a novel genome browser. This genome browser supports mainstream file formats, optimizes the display of genome-mapped features, and allows users to quickly navigate and accurately locate specific genes or genomic regions. Users can view detailed information on specific features and perform in-depth analyses with ease. The genome browser of SGS offers a rich array of track visualizations, including gene structure, genome variation loci, epigenetic signals (such as chromatin accessibility signals, differential peaks, gene-CRE links, methylation signals, chromatin interactions, and sc-eQTL loci information), and more (**Figure 1**). SGS not only supports data scaling and filtering but also provides a variety of statistical methods, as well as flexible track splitting and grouping settings, ensuring that users can deeply explore and analyze data in detail according to their needs.

Figure 1. Screenshot of genome tracks in the SGS genome browser framework. Gene structure, chromatin accessibility signal, Gene-CRE link, sc-eQTL loci, chromatin interaction, methylation loci signal, and genomic variant feature are displayed from top to bottom.

Furthermore, SGS's dual-chromosome visualization module provides a powerful visual comparison of single-cell and spatial epigenomic multimodal signals (**Figure 2**). By utilizing a double chromosome display strategy, this module presents several advantages. Firstly, the top and bottom coordinates can cover different genomic regions, enabling the visualization of single-cell long-range interactions. Secondly, users have the flexibility to independently shift or zoom these coordinate regions. This capability facilitates the comparative visualization of cell type-specific epigenomic signal differences in multiple regions simultaneously, enhancing our understanding of cell-specific regulatory patterns.

(2) Deep Integration and Interaction of Genome Browser Panel and Single-Cell Panel Although both SGS and Vitesce can integrate and display annotation information of single cell and genome-mapped signals on the same panel, the interaction design between single cell and genome panels of Vitesce is weak, and the association query of specific genes between a single cell panel and the genome interface cannot be performed.

In contrast, SGS utilizes genome features (such as peaks, genes, and sc-eQTL) as interactive anchors to achieve coordinate visualization between the single-cell panel and the genome browser. Users can click on marker genes or peaks to simultaneously view their dynamic changes at the single-cell level and navigate to the genome browser to observe variations in epigenetic signals and provide detailed annotations of these features. For single-cell eQTL data, SGS allows users to click on marker genes in

the sc-eGenes table to view the gene's expression changes, while highlighting the associated sc-eQTL loci in the genome browser. The rich track information display of the genome browser and the interactive design between the single-cell and genome browser enable SGS to better handle the visualization and integrated exploration of complex epigenetic multimodal data, such as sn-m3C-seq data, which include DNA methylation and 3D chromatin conformation, and sc-eQTL data.

(i) Example1: Visualization of Single-Cell eQTL Data

The SGS provides researchers with a powerful tool to visually explore sc-eQTL (single-cell expression quantitative trait loci) data, providing valuable insights into the impact of gene regulation on cellular function and disease mechanisms. It presents sc-eQTL data from the OneK1K research, including 982 individuals. The genome browser presents multiple GWAS tracks of sc-eQTLs across various cell types. Users can easily locate specific sc-eQTL regions by entering gene symbols (e.g. *BLK*) or genomic positions (e.g. chr8:11300703-11611527) (**Figure 3a, b**). The single-cell visualization component displays UMAP of 14 major cell types, whereas the interactive table at the bottom lists sc-eGenes (genes significantly associated with sc-eQTLs) related to specific cell types. Clicking on a gene in the marker table allows users to examine its expression patterns across cell types and navigate to the associated sc-eQTL region in the genome browser.

By clicking on the "*BLK*" gene in the marker table allows users to explore its expression heterogeneity across cell types and navigate to the genomic region in the genome browser (**Figure 3d**). In the single-cell panel, we can observe its specific high expression in Bmem cells (**Figure 3c**). The genome browser panel highlights *BLK*-specific sc-eQTLs in red, with other sc-eQTLs in gray. Users can access details such as the *BLK* sc-eQTL site ID, p-value, etc. Here, we observed that many eSNPs associated with the "*BLK*" gene show correlations with *BLK* expression in CD4 NC, CD8 ET, CD8 NC, B Mem, and B IN cells. Notably, among the associated eQTL loci, the rs2736336 locus was identified. Previous studies have reported that the rs2736336 variant leads to differential expression of *BLK* in B Mem cells, indicating its potential role in inter-individual variability of B lymphocyte tolerance.

Figure 3. The screenshot of OneK1K sc-eQTL visualization demo. The panel consists of two components: the genome framework and the single-cell component. The genome framework provides insights into the structure of the *BLK* gene region (a) and highlights cell type-specific sc-eQTL loci. The significance of each loci is represented by the size and height of the points, with sc-eQTLs associated with the "*BLK*" gene highlighted by red circles (b). The single-cell component displays the UMAP of 14 cell clusters and the expression pattern of the *BLK* gene across different cell populations (c). The marker table displays essential information such as the p-value, q-value, log fold change (logFC), and other relevant details of the marker gene (d).

(ii) Example2: Single-nucleus Methyl-3C Sequencing (sn-m3C-seq) Data Visualization

As the first example, we show how SGS enables the synchronized visualization of Single-nucleus Methyl-3C Sequencing (sn-m3C-seq) data encompassing DNA methylation and 3D chromatin conformation during the development of the human frontal cortex (PFC) and hippocampus (HPC)¹. Figure 4A displays two different views that integrate features selected in the SG visualization mode: (1) The genome browser displays the gene structure of *RORB* gene positioned near the region of rs500102 snp (Figure 4a). Below this view, the panel showcase the CG methylation signals alongside Hic track of specific cell types (e.g., PFC 2T RG-1; PFC adult L1-3 NRXN2; PFC adult L4-5 FOXP2 etc) (Figure 4b). (2) The single-cell panel showcases the cell atlas of sn-m3C-seq data obtained from 13 developmental adult PFC and 9 HPC samples (Figure 4c), which provides an overview of the distribution of 10 primary cell types and CG methylation patterns of *RORB* gene. Users can select a key feature in the single-cell marker table to directly navigate to the corresponding region in the genome browser to obtain multi-omics layer information through the adaptive communication design. Of particular interest, by clicking on the *RORB* gene within

the marker gene table, users can navigate to the corresponding region in the genome browser. This allows for the observation of the noticeable enhanced chromatin interaction strength specifically in the adult PFC L4-5 FOXP2 cell population within the *RORB* region, accompanied by a decreased CG methylation signal compared to other cell populations. Alternatively, in the single cell panel, users can investigate the heterogeneity in CG methylation signal distribution among various cell populations. Notably, a decrease is observed especially in excitatory neurons within the PFC L4-5 FOXP2 cell cluster, which is consistent with previous research findings (Figure 4d). In this case, we demonstrate the powerful capabilities of SGS in visualizing complex epigenomic multimodal data. Through multi-panel integration and adaptive communication, SGS enables the integrated visualization of epigenome genome-mapped signals (gene annotation, DNA methylation, 3D chromatin conformation), and single-cell methylation information within a unified interface.

Figure 4. The main panel illustrates the single-cell epigenomic multi-omics data (sn-m3C-seq) capturing DNA methylation and 3D chromatin conformation during human frontal cortex (PFC) development. The left genome browser displays genome annotations in the region near the *RORB* gene, along with HiC and CG methylation signals for specific cell types: 2TRG-1, adult L1-3 NRXN2, adult L4-5 FOXP2, 2T eMGE, adult MGE-ERBB4, and adult ODC (a, b). The single-cell panel on the right presents the cellular profile of the PFC and a scatter plot showing the expression of the *RORB* gene (c, d).

(3) 3D Spatially Resolved Transcriptomics Data Visualization

The emergence of organ-level and organism-level 3D spatially resolved transcriptomics (SRT) provides more comprehensive insights into biological principles compared to conventional 2D slices. At present, Vitesce only supports 3D multimodal mass spectrometry imaging dataset visualization, which cannot display 3D SRT data. One of the significant advancements of SGS over Vitesce in the realm of 3D transcriptomic data visualization based on continuous 2D virtual slice reconstruction results. SGS offers advanced 3D transcriptomic data visualization capabilities that enhance the understanding of gene expression patterns within a three-dimensional context, providing deeper insights into spatial relationships. Users can engage in interactive 3D data exploration with surface model plots, visualizing each gene's spatial expression pattern through color opacity that reflects expression strength. Furthermore, users can investigate gene spatial co-expression, examining co-occurrence patterns (**Figure 5**). It also facilitates the observation of overall changes in cell populations and comparative morphological analysis. The integration of single-cell and genome data for a comprehensive analysis of the relationship between 3D transcriptomics and other multimodal data.

(4) Enhanced Accessibility and Collaboration

SGS is a cross-platform GUI visualization software with a client-server separation, offering a graphical one-click installation, a visual interface for settings, and database querying capabilities. Its design philosophy focuses on providing a rapid and user-friendly visualization platform for non-programmers, emphasizing team collaboration and real-time data sharing. In contrast, Vitesce is a flexible and extensible web-based framework that partially overcomes the server dependency but limits data retrieval queries and fast navigation. Vitesce is primarily geared towards users with some programming background, allowing for personalized customization of visualization portals through the configuration of visualization components.

(i) Interface Operability

A notable distinction in terms of interface operability is that Vitesce relies on configuration files to customize interface settings, including color schemes, image transparency, and the display of cell labels. While this approach offers flexibility, it necessitates a certain level of understanding of the configuration parameters, which can elevate the barrier to entry and diminish the interface's operability and user-friendliness. In contrast, SGS provides a more intuitive graphical installation and graphical setup. Users can perform operations such as theme setting, single-cell label configuration, and track style settings through a graphical user interface (GUI), without the need to directly edit configuration files (**Figure 6**). This method significantly reduces the learning curve for users and enhances the interface's operability.

Figure 6. Interface functional description of SGS.

(ii) Collaborative Data Visualization

In addition, another major advancement of SGS over Vitesce is that it allows collaborative data visualization and management. Team collaboration is a much-needed feature in the exploration of single-cell and spatial multi-omics visualization. We have fixed and improved the SGS collaboration visualization feature, which allows multi-user real-time collaboration and editing, document comments, cell type annotation, access management and data management (**Figure 7**). The real-time collaboration and editing function allows multiple users to work simultaneously on the same project. This means team members can see each other's changes in real-time, thereby accelerating decision-making processes and project completion speeds. Document comments and discussions allow users add comments and initiate discussions within data visualizations, aiding team members in communicating and providing feedback on specific content.

Figure 7 . Collaborative Visualization Features of SGS. (A) Genome -mapped Data Collaboration. SGS supports multiple users in adding comments to features (a), viewing collaboration history (b), and renaming features (c) for collaborative visualization. (B) Collaborative Cell Annotation. SGS enables multiple users to create annotation groups (a); and compare different users' annotation results, such as results from User 1 and User 2 (b, c).

(iii) Data and User Management

In term of data management, Vitesce lacks a dedicated data management system, often requiring the creation of individual configuration files and customized visualization portals for each research project. While, SGS is designed with the needs of multi-project and multi-species research, featuring a robust data management system. It empowers users to manage multiple projects effectively, offering capabilities for batch addition, deletion, and data grouping to enhance both efficiency and flexibility (Figure 8A). In addition, SGS offers user management capabilities, supporting operations such as adding, deleting, and setting permissions for users. In contrast, Vitesce does not have a user management system (Figure 8B).

In short, SGS improves the efficiency of team collaboration by providing real-time collaboration, document review, discussion, permission and data management. By leveraging these capabilities, teams can complete projects more efficiently and achieve better collaborative outcomes.

2. Manuscript Writing Adjustments

(1) Introduction Section: We have revised the overview of the limitations of existing visualization tools in the introduction (line 63-69) .

(2) Overview of SGS Section Added: A new section has been added to elucidate the core advantages of SGS (line 90-115).

(3) Graphical Installation and Collaborative Visualization Section: We have included examples of the simplified installation process of SGS (Figure S4) and illustrations of its multi-user collaboration features (Figure S5) in the supplementary files.

(4) A Comparison With Existing Visualization Tools Section Added: we have add a new section dedicated to comparing SGS with existing visualization tools. This comparative analysis has been added to the supplementary material (line 1-41). Furthermore, to illustrate the differences, we have visualized the pbm5k scATAC data using both Vitesse and SGS (**Figure S1**). Lastly, we have made revisions to Table S1 to enhance clarity and comparison.

(5) User Case Section Revised: We have expanded the user case section with examples such as single-cell eQTL data visualization, ME11 Spatial-ATAC-seq visualization, and 3D spatial transcriptome data visualization to fully demonstrate the multimodal data visualization potential of SGS (line 502-635).

In summary, SGS provides a unified interface for visualizing scMulti-omics data, including genomics, transcriptomics, proteomics, and epigenomics data with single-cell or spatial resolution. This comprehensive visualization browser aids in systematically interpreting cellular heterogeneity, tissue organization, and biological processes. We hope that these revisions and additions will adequately address your concerns and highlight the powerful capabilities of SGS in single-cell and spatial multimodal data visualization, as well as its advantages over existing visualization tools like Vitesse. We trust that these changes will sufficiently alleviate your reservations.

2. In Supplementary Table 1, the only differences between SGS and Vitesse seem to be the session URL sharing function and the accepted data formats. These differences alone may not be sufficient to justify the publication of SGS in a top journal.

Response: Thank you for your thorough review and valuable feedback on our manuscript. We acknowledge that the initial submission did not sufficiently highlight the functional differences between SGS and Vitesse. In the revised manuscript, we have supplemented the Table S1 and provided a detailed comparison of the two tools in terms of visualization capabilities, interactivity, view coordination, user collaboration, and data formats to clearly demonstrate the unique value and innovations of SGS. Here is a description of the content in Table S1:

(1) Visualization Functions

While SGS and Vitessce share some similarities in functionality, SGS is specifically tailored as an interactive browser for single-cell and spatial epigenomic multimodal data visualization. SGS not only supports foundational visualization features such as spatial feature visualization, multi-feature comparison, multi-slice visualization, and chromatin accessibility signals but also offers an extensive range of track visualization functions. These include gene structures, genome variation sites, epigenetic signals (e.g., chromatin accessibility signals, differential peaks, gene-CRE links, methylation signals, chromatin interactions, and sc-eQTL loci information). On the other hand, Vitessce primarily focused on the visualization of expression-based single-cell and spatial multimodal data. It has limited capabilities in displaying genome-mapped signals such as gene structures and chromatin accessibility signals. Furthermore, it lacks precise genomic navigation and querying capabilities. In terms of 3D transcriptomic data visualization, Vitessce lacks support for 3D SRT visualization. In contrast, SGS provides sophisticated 3D SRT data visualization tools. Users can delve into interactive 3D data exploration, employing surface model plots and gene spatial expression scores to vividly depict the spatial expression patterns of each gene. The interactive 3D visualization bridges the gap between researchers and spatially resolved transcriptomics, thereby accelerating studies related to processes such as embryogenesis and organogenesis.

(2) Interactivity

SGS has been thoughtfully designed with user experience and interface interactivity in mind, providing an intuitive interface and smooth operational logic. Beyond the core functionalities found in mainstream tools like Vitessce (such as Cell selection and export, Move/Zoom, and Image setting), SGS also supports data scaling, filtering, and various statistical methods, as well as flexible track splitting and grouping settings. Unlike Vitessce, which relies on configuration files, SGS simplifies installation, theme setting, cell label switching, and track grouping through a graphical user interface, reducing the learning curve for users and enhancing the operability of the interface.

(3) Views Coordination

In terms of view coordination, SGS utilizes genomic features (such as peaks, genes, and eQTLs) as interactive anchors to synchronize single-cell panels with genome browser coordinate visualization, whereas Vitessce has limited capabilities in interaction between single-cell and genome browser and lacks precise navigation and querying functions. SGS allows users to view the dynamic changes of genes or peaks at the single-cell level while quickly navigating to the genome browser to explore changes in epigenomics signals and detailed annotations. For single-cell eQTL data, users can click on marked genes in the sc-eGenes table to view gene expression changes and highlight related sc-eQTL loci in the genome browser. SGS's rich track information and effective interaction design between single-cell and genome browsers enable more efficient handling and comprehensive exploration of complex epigenomic multimodal data visualization.

(4) User Collaboration

A significant advancement of SGS over Vitesce is its support for collaboration among multiple users. SGS enables team members to efficiently share, edit, and manage data, allowing multiple users to work on a project simultaneously and view each other's modifications, thereby accelerating decision-making and project completion. Additionally, users can add comments and initiate discussions within data visualizations, fostering communication and feedback among team members.

(5) Data Format

Lastly, in terms of data formats, SGS supports a variety of popular multimodal data formats (such as h5ad, h5mu, MethyC, VCF, Longrange, etc.). This compatibility makes SGS more attractive and responsive to the needs of a broader range of users, especially in the context of rapidly evolving single-cell research with new data formats emerging. Furthermore, SGS offers flexible plugins and SgsAnnData R packages for format conversion, further simplifying the user's workflow.

We hope that these additional explanations sufficiently clarify the uniqueness and significance of SGS and addressed your concerns. Thank you for your review and valuable feedback.

	Comparison item	SGS	Vitesse	AtlasXplore	Cellxgene	Loom-Viewer	Single Cell Explorer	UCSC Cell Browser	Loupe Browser	ST Viewer	TissUUmapi3
Visualization Function	Spatial feature visualization	✓	✓	✓	✓	✓			✓	✓	✓
	Highlight gene expression	✓	✓	✓	✓	✓	✓	✓	✓	✓	✓
	Highlight metadata	✓	✓	✓	✓	✓	✓	✓	✓	✓	✓
	Tissue image visualization	✓	✓		✓				✓	✓	✓
	Multi-feature comparison	✓	✓	✓	✓	✓		✓	✓		
	Multi-slice visualization	✓	✓		✓				✓		✓
	Multiple embeddings	✓	✓	✓	✓	✓	✓	✓	✓	✓	✓
	Gene structure visualization	✓	✓	✓							
	Chromatin accessibility signal	✓	✓	✓							
	Meta table	✓	✓	✓	✓						
	3D transcriptomic data visualization	✓									
	Feature sequence visualization	✓									
	Track information	✓									
	sc-eQTL loci visualization	✓									
	Methylation loci Visualization	✓									
	Gene-CRE link visualization	✓									
	Motif logo visualization	✓									
	Variant feature visualization	✓									
Chromatin interaction visualization	✓										
Dual-chromosome comparative visualization	✓										
Interactivity	Cell selection and export	✓	✓	✓	✓		✓	✓	✓	✓	✓
	Move/Zoom	✓	✓	✓	✓		✓	✓	✓	✓	✓
	Image setting	✓	✓		✓				✓	✓	✓
	Global theme setting	✓									
	Track Feature search (gene/snp/region)	✓		✓							
	Track scale/statistics	✓									
	Graphical installation	✓							✓	✓	
Views coordinations	Single-cell views coordinations	✓	✓	✓				✓	✓		
	Single-cell view and genome browser coordination	✓		✓							
User Collaboration	Web Sharing	✓	✓	✓	✓	✓	✓	✓			✓
	Session URL sharing	✓			✓			✓			
	Create and manage projects	✓									
	Add and delete data in batches	✓									
	Data organization and classification	✓									
	Track comments	✓									
	Real-time annotation	✓					✓				
	Authority management	✓									
Data Format	anndata/anndata.zarr	✓	✓		✓		✓				✓
	mudata/mudata.zarr	✓	✓								
	GFF	✓									
	VCF	✓									
	BED	✓	✓	✓							
	BigWig	✓	✓	✓							
	MethylC	✓									
	Longrange	✓									
	BigInteract	✓									
	HIC	✓									
GWAS	✓										

Table S1: Comparison of SGS with existing mainstream single-cell and spatial multi-omics visualization tools. This table contrasts the capabilities of SGS with those of Vitesse, AtlasXplore, Cellxgene, Loom-Viewer, Single Cell Explorer, UCSC Cell Browser, Loupe Browser, ST Viewer, TissUUmapi3. The light yellow section primarily highlights the fundamental visualization features such as Spatial feature visualization, highlight gene expression, Multi-feature comparison, genome-mapped data visualization etc. The light orange section focuses on interactivity features including cell selection/export, Move/Zoom, Image setting, and Track setting etc. The light blue section focuses on views coordination features including single-cell views coordination, single-cell panel and genome browser coordination. The light purple section focuses on user collaboration features including web sharing, data or project management, real-time collaboration etc. The light green section compares data format in terms of loading anndata, mudata and genome track files.

3. Based on the description in the manuscript, the association between the SC and SG modules appears weak. It would be much more useful if users could select certain cells in the SC module and display the epigenomic information for those selected cells in the SG module. Otherwise, users could simply open two visualizers side-by-side (e.g., Loupe Browser for single-cell data and UCSC Genome Browser for epigenomic data).

Response: We greatly appreciate your insightful suggestion regarding enhancing the interaction between the single-cell panel and the genome browser panel. The functionality you propose—selecting specific cells within the SC module and displaying the corresponding epigenomic information for those cells in the SG module is indeed invaluable to our users.

Currently, to address the visualization of complex epigenomic multimodal data, SGS integrates single-cell and genome browser visualization panels, establishing effective associations through key features such as genes, peaks, eQTLs, and DMRs, enabling coordinated visualization across both panels. When users interact with these features, the panels automatically query, navigate, and dynamically render the changes of these features at both the genomic and single-cell levels.

For instance, with scRNA/scATAC data, users can click on marker genes or peaks to view their dynamic changes at the single-cell level and simultaneously navigate to the genome browser to observe variations in epigenetic signals and detailed annotations of these features. For single-cell eQTL data, SGS allows users to click on marker genes in the sc-eGenes table to view gene expression changes, while highlighting the associated sc-eQTL loci in the genome browser. Moreover, SGS supports the visualization of epigenetic signals generated from specific cell populations using analytical tools like ArchR and snapATAC, such as chromatin accessibility, DNA methylation, and chromatin interactions.

Given that the significant value of such a feature, particularly when dealing with the complex and vast datasets characteristic of single-cell and spatial epigenomic multimodal data. These datasets encompass high-dimensional, noisy, and sparse information, such as histone modification, DNA methylation, chromatin accessibility, and chromatin interactions. The scale of these datasets is often enormous, involving the processing of millions of cellular epigenetic signal files, such as fragments.tsv files in scATAC analysis, .allc files in single-cell DNA Methylation analysis, and .cool files in scHiC analysis. Achieving real-time selection of cell sets and dynamic rendering of epigenomic modification signals imposes significant computational overhead, especially when dealing with single-cell HiC and methylation data, which further increases response times and limits rapid data visualization, which is currently available with few visualization tools.

Through the integrated and interactive design of the single-cell and genome browser visualization panels, we enable users to quickly navigate and integrate data visualization between the two modules within the same interface. This design eliminates the need to open two separate browser windows and avoids the inconvenience of switching between multiple interfaces. This approach holds potential for uncovering intricate relationships and interactions among diverse types of molecular information within distinct omics layers. We hope these additional explanations have addressed your concerns.

4. For single-cell multi-omics data that profile both gene expression and chromatin accessibility in the same cells, can SGS display gene-CRE linkages?

Response: Thank you for the valuable feedback you have shared. Regarding your specific inquiry about the SGS's capacity to display gene-CRE linkages. In response to your question, we are pleased to confirm that SGS is indeed capable of displaying gene-CRE linkages. Specifically, Figure 7 displays the human hematopoietic scATAC dataset. The genome browser shows genome tracks for the *VSTM1* gene, including gene structure and chromatin accessibility signals for PreB, Mono, CD4 N cells, and other cell types (**Figure 7a**). The single-cell panel showcases a split-screen layout, displaying both the cell embedding plot and the distribution *VSTM1* gene expression (**Figure 7d**). By clicking on the *VSTM1* gene in the single-cell panel's marker table (**Figure 7f**), users can immediately observe their expression patterns across different cell types (**Figure 7e**) and navigate to the gene region to explore the distribution of epigenomic modification signals (**Figure 7(b, c)**), such as chromatin accessibility and peak-to-gene links, in the genome browser framework. This mechanism facilitates synchronous exploration of identical data types with linked views, allowing multimodal datasets to be displayed across different browser panels.

Figure 7. The SG visualization mode presents human hematopoietic scATAC datasets, comprising the genome framework (left) and single-cell components (right) (a, d). The genome browser displays tracks for the VSTM1 gene, including gene structure and chromatin accessibility signals across various cell types, as well as co-accessibility links between peaks and genes (b, c). The single-cell panel features a split-screen view with a cell embedding plot and the distribution of VSTM1 gene expression (e). Clicking on marker genes in the table enables the exploration of gene activity scores and genome-mapped signals (f).

Furthermore, we have revised the section of our manuscript relating to SGS's capabilities in visualizing epigenomic multimodal data. We have clarified that **"This framework enables a rich array of track visualizations, including gene structures, genome variation loci, epigenetic signals (such as chromatin accessibility signals, differential peaks, gene-CRE links, methylation signals, chromatin interactions, and sc-eQTL loci), and more (Figure S7). Additionally, it not only supports mainstream file formats and commonly used operations but also provides innovative features for the integrative analysis of genome-mapped signals. These features include an optimized feature display, dual chromosome visualization, and efficient gene or region navigation."** We believe that these additional explanations and revisions comprehensively address your concerns and enhance the understanding of SGS's potential in visualizing single-cell and spatial epigenomic multimodal data. The revised analyses and results have been incorporated into the manuscript, further strengthening the demonstration of SGS's capabilities.

5. Can SGS input data already processed by other software, such as Seurat and Scanpy?

Response: Thank you for your insightful inquiry about the compatibility of the SGS with data that has been processed using other software platforms, such as Seurat and Scanpy. We appreciate the importance of this functionality and are pleased to provide a detailed response.

Currently, SGS supports the direct visualization of h5ad files, which are the output of Scanpy. For data that has been processed with Seurat, user need to converted the object to h5ad for SGS visualization. The SgsAnnData R package offers a robust solution for transforming analysis results from a multitude of tools, including Seurat, ArchR, Signac, and Giotto, into the h5ad format that SGS can recognize (**Figure 8A**). Users can use the SeuratToAnndata function to convert Seurat objects into h5ad files. This transformation requires inputting parameters such as object, outpath, assays, groups, reductions etc (**Figure 8B**). Post-conversion, users can effortlessly select the corresponding h5ad file for immediate visualization.

To ensure that users can fully exploit this feature, we have supplemented the documentation of SGS with a comprehensive guide detailing the conversion process. We are confident that these enhancements will significantly streamline the process of importing existing analysis results into SGS and conducting efficient data visualization.

Figure 8. (A) SgsAnnData of SGS used for the conversion of single -cell analysis objects from tools like Seurat, Giotto, Signac, and ArchR into the AnnData format. (**B**) The SeuratToAnndata function converts Seurat objects into h5ad files, with parameters including assay, metadata, reduction, outpath, etc.

6. Without watching the tutorial videos, it is quite difficult to begin using SGS. For instance, it seems that users must first deploy SGS before loading data and performing analyses or visualizations. This process seems tedious, and some users may feel intimidated and stop at this point. I wonder if there is a way to simplify the process of initializing an analysis?

Response: We greatly appreciate your valuable suggestion on the accessibility of the SGS, particularly regarding the challenges users may face during the initial software deployment and analysis initiation phases. Your insights are crucial and have prompted us to further enhance the user experience of SGS. Inspired by tools like xshell or finalshell, our software adopts a client-server separation strategy to rapidly deploy a collaborative visualization and project management platform for single-cell and spatial multimodal data, tailored for research teams. Leveraging Docker and Flutter technologies, we have overcome complex configurations and environmental dependencies while ensuring compatibility across Linux, Windows, and MacOS. This design enables convenient management and collaborative visualization of multiple research project data on a single server or different servers for single-cell and spatial multimodal research projects, facilitating collaborative research efforts. Acknowledging the potential challenges users may face with SGS's usage model compared to standalone or web-based visualization software, we have further simplified the installation process based on your valuable suggestions.

To make the initial installation process more user-friendly, we have implemented the following improvements to the installation and deployment steps of SGS:

(1) Simplified Installation Process

In our latest version of SGS, we have made it a priority to simplify the deployment process to minimize the initial challenges users face. We have refined the installation process to be more user-friendly, allowing users to directly access the installation interface by downloading and clicking on the SGS software. With just a single click, users can initiate the installation process by providing essential information such as server IP and password (**Figure 9**).

(2) Updated Tutorials and Documentation

We have updated the installation tutorials and detailed usage documentation to provide clearer and more comprehensive guidance. These resources will help users become familiar with SGS operations more rapidly and reduce confusion during installation and use. The documents can be viewed at the link below: <https://sgs.bioinfotoolkits.net/document/installation.html>.

We are hope that these enhancements will improve the ease of use of SGS, allowing a broader range of users to effortlessly begin leveraging SGS for data analysis and visualization.

Reviewer #2:

The paper presents the SGS Genome Browser, a novel tool designed to facilitate the integrative and collaborative exploration of single-cell and spatial multimodal data. With the growing complexity of data generated from single-cell and spatial technologies, there is a need for tools that can help researchers visualize, analyze, and compare features from multiple modalities (e.g., RNA expression, protein levels, DNA accessibility, etc).

However, there are parts that could be clarified to strengthen the manuscript:

1. While collaboration is mentioned as a core feature, there are limited specific details or examples showing how real-time collaboration works, especially in terms of data security, version control, or simultaneous editing. It will be helpful to provide demonstrations of how collaboration is implemented (e.g., how users can annotate the same dataset in real-time, how conflicts are resolved, etc.), along with information on security and user permissions in collaborative settings.

Response: We greatly appreciate your insightful comments on the collaborative features of SGS. Your emphasis on the need for specific details and examples of real-time collaboration, particularly regarding data security, version control, and simultaneous editing, is well-taken. In response to your suggestions, we have made the following revisions:

1.Fixed and improved SGS collaborative visualization features:

In the latest version of SGS, we have fixed SGS collaborative visualization related bugs. At present, SGS collaborative visualization functions mainly include the following aspects:

(1) Real-time Collaboration and Editing: SGS now supports real-time online collaboration, enabling multiple users to work simultaneously on the same project. This feature ensures that team members can see each other's changes instantly, thereby accelerating decision-making and enhancing project completion speeds.

(2) Document Comments and Annotations: We have introduced comment and annotation features within data visualization, allowing users to communicate and provide feedback on specific content, including annotating features of the same dataset, which enhances communication and collaboration among team members.

(3) Access Control Management: SGS allows the initiator of data sharing to set different access permission levels (administrator, common user), ensuring that only authorized team members can view or edit data, thus safeguarding data security.

2.Demonstration of Collaboration in the Manuscript:

To provide a more intuitive demonstration of SGS's collaborative features, we have included a specific case example in the revised supplementary manuscript (line 221-257). This example illustrates how multiple users can collaborate in real-time to visualize genome-mapped data. They can perform operations such as adding synchronized comments and annotations on genes or peaks (**Figure 10A**). Additionally, we have showcased how multiple users can annotate the same single-cell dataset (**Figure 10B**).

2. The paper may underplay the challenges of multimodal data integration, such as dealing with batch effects, cross-platform normalization, or differences in resolution between spatial and single-cell data. It will be helpful to address these challenges more explicitly, discussing how SGS deals with such issues, or include future directions for tackling these common obstacles in multimodal data analysis.

Response: We sincerely appreciate your review of the multimodal data integration challenges within the SGS manuscript. We fully agree that addressing issues such as batch effects, cross-platform normalization, and resolution differences between spatial and single-cell data is critical to achieving accurate data interpretation. Below is our discussion of these challenges and how SGS is addressing them, including our future research directions and plans.

(1) Batch Effects and Cross-platform Normalization

Batch effects are indeed a significant concern in multimodal data integration, arising from technical variations unrelated to the biological variables of interest. These variations are introduced into high-throughput data due to changes in experimental conditions over time, use of different labs or machines, or different analysis pipelines²⁻⁵. In multimodal data, these effects are more complex due to the involvement of multiple types of data measured on different platforms with varying distributions and scales^{6,7}. These batch effects can lead to incorrect conclusions and are a paramount factor contributing to irreproducibility. To address batch effects, researchers employ a variety of strategies, including Location-Scale (LS) methods (like ComBat^{8,9}, reComBat¹⁰), Matrix-Factorization (MF) methods (like SVA¹¹, RUVseq¹²), Distance-Neighborhood (DN) methods (like mnnCorrect¹³, deepMNN¹⁴), Linear Embedding Models (harmony) and Deep-Learning (DL) methods (like AutoClass¹⁵, DESC¹⁶, scGen¹⁷, scVI¹⁸).

Despite the existence of many batch-effect correction algorithms (BECAs) have been proposed, batch effect correction is still an active area of research. There are still many challenges in this area. One of the major challenges of batch effect issues is evaluation and quantifying the impact of batch effects on the data. The development of methods capable of precisely identifying and quantifying batch effects is essential to minimize their impact on the downstream data analysis¹⁹. Another significant challenge lies in the generalizability of BECAs across various datasets and experimental conditions. It is essential to develop methods that can generalize well across different datasets and experimental conditions to ensure the reliability and reproducibility of the data. The third challenge is the selection of software and algorithms, with new methods such as CellANOVA²⁰, scMerge2²¹, scDisInFact²², and SPEEDI²³ emerging and being evaluated.

(2) Differences in Resolution Between Spatial and Single-cell Data

Integrating single-cell and spatial transcriptomic data presents challenges such as data noise, heterogeneous data modalities, and disparities in spatial resolution. Addressing these challenges involves several aspects:

(i) Integration of Heterogeneous Data Modalities

Constructing a coherent picture of the tissue under study requires the spatially aware integration of heterogeneous data. This integration is challenging due to the vast differences in feature counts across modalities (e.g., the number of proteins versus transcripts measured) and their distinct statistical distributions. The challenge is further compounded when integrating spatial information with feature counts within each modality²⁴. Heterogeneous omics combinations from different studies, which may include one or multiple missing modalities, create a mosaic-like dataset. Integration methods must reconcile batch heterogeneity and technical differences, enabling modal imputation and batch correction for downstream analysis. To address these issues, horizontal integration algorithms can correct technical variations between different parts of the same or different samples before applying vertical integration²⁵. For instance, the deep learning-based linear deep generative model, gimVI, interpolates missing values and integrates spatial transcriptomics (ST) and single-cell RNA sequencing (scRNA-seq) data by training a deep model²⁶. Alternatively, the Multimodal Intersection Analysis (MIA) approach, akin to enrichment analysis principles, integrates scRNA-seq and ST sequencing data to deduce the enrichment degree of cell subpopulations within tissue space by assessing the correlation of specific genes across both data types²⁷. Tangram, a mapping approach based on a deep learning framework, enables the integrated analysis of diverse ST data and single-cell or scRNA-seq data, effectively assigning the most probable cell type to each spot within the spatial data²⁸.

(ii) Integration of Spatial Information with Varying Resolutions

There is a significant disparity in resolution among different spatial omics technologies such as 10x Genomics Visium, Stereo-seq²⁹, 10x Genomics Xenium, MERFISH³⁰, and seqFISH³¹, posing a challenge for data integration. The inconsistency in resolution means that cellular and molecular information captured by different technologies may not correspond directly on a spatial scale. Current integration methods for scRNA-seq and spatial transcriptomics can be broadly categorized into two approaches: deconvolution and mapping³². Deconvolution methods involve constructing mathematical or statistical inference models where scRNA-seq data serves as background data. These methods combine spatial transcriptomics sequencing data with known cell types or marker gene expression patterns from scRNA-seq data to infer cell types at each spot. Among deconvolution methods, Cell2location³³, a probability model-based deconvolution integration method, uses a Bayesian statistical model to jointly analyze scRNA-seq and spatial transcriptomics data. This model demonstrates a strong ability to identify fine-grained cell types within complex tissues and is highly sensitive to subtle cell type variations. In contrast, Stereoscope³⁴, a probability model-based deconvolution method, can deconvolve spatial transcriptomics data using complete expression profiles rather than relying on specific gene markers. On the other hand, mapping aims to establish a correspondence between scRNA-seq and spatial transcriptomics sequencing data, aligning and mapping them within the spatial domain. This approach enables the spatial visualization and analysis of cell types or gene expression patterns. Mapping methods such as LIGER³⁵, Tangram²⁸, stPlus³⁶, and CeLEry³⁷ can map cell types based on scRNA-seq data onto spatial datasets, achieving spatial visualization and analysis of cell types or gene expression patterns.

(3) Future Research Directions

Looking ahead, SGS is poised to employ a comprehensive suite of integration strategies to address the intricacies of multimodal data integration. These include vertical, horizontal, and mosaic integration algorithms, each tailored to address specific challenges³⁸:

(i) Vertical integration algorithms

These algorithms are used to integrate different omics modalities measured on the same sample or cell. These algorithms are tailored to associate various types of omics data, such as gene expression and protein abundance or chromatin accessibility, which are termed vertical integration. Our focus will be on algorithms like Seurat³⁹ and MOJITOO⁴⁰, which have demonstrated exceptional performance in merging RNA expression with protein abundance. Additionally, scAI⁴¹ and MOJITOO have shown promise in integrating RNA expression with chromatin accessibility, a critical aspect of vertical integration.

(ii) Horizontal integration algorithms

These algorithms facilitate the correction of batch effects across multi-omics datasets. This type of integration is crucial when dealing with data from different experiments or batches that need to be combined while preserving biological variation. We have identified totalVI⁴² as a superior algorithm for integrating single-cell RNA + protein datasets, and UINMF⁴³ for its excellence in handling single-cell RNA + ATAC data. These algorithms excel in horizontal integration by effectively mitigating batch effects and maintaining the integrity of biological signals.

(iii) Mosaic integration algorithms

These algorithms are designed to integrate single-cell datasets that share at least one type of omics information. This process is known as mosaic integration and is particularly useful when dealing with datasets that have some overlap in the omics data they contain. We have highlighted the efficacy of algorithms such as totalVI, UINMF, and scArches⁴⁴ in mosaic integration, given their ability to manage datasets with both shared and unshared features, thus allowing for a more flexible and comprehensive integration of diverse single-cell multi-omics datasets.

However, it is important to note that each type of integration tool has its own strengths and limitations, and the choice of method depends on the nature of the data, the sources of batch effects, and the specific goals of the analysis. Therefore, we will carefully evaluate the performance of different integration tools in each specific context based on the latest research and methodological reviews before choosing one for analysis.

We have also revised the manuscript's discussion. The revised text is as follows. **"To further enhance the exploration of high-dimensional data modalities and address the underlining the challenges of multimodal data integration, like batch effects, cross-platform normalization, data modality heterogeneous, differences in resolution between spatial and single-cell data etc. SGS is poised to employ a comprehensive suite of integration strategies to address the intricacies of multimodal data integration. These include vertical integration methods (like MOJITOO and scAI), horizontal integration methods (like totalVI and UINMF), and mosaic integration algorithms (like totalVI, UINMF, and scArches), each tailored to address specific challenges."** We hope this detailed outline

of our future research directions addresses your concerns and provides a clear vision of how SGS intends to tackle the common obstacles in multimodal data analysis.

3. In the conclusion section, authors mentioned: "SGS plays a crucial role in advancing our understanding of differentiation trajectories, the underlying gene regulatory networks, cell-to-cell interactions, microenvironmental spatial organization, cellular lineages, and clonal dynamics". It is not clear whether trajectory analysis, regulatory network, or cell-to-cell interactions are supported by SGS, in terms of visualization.

Response: We sincerely appreciate your valuable feedback on our discussion section, particularly your observations regarding the ambiguity surrounding the types of analyses and visualizations supported by the SGS. Your comments have prompted us to clarify the functionalities and application values of SGS with greater precision.

We acknowledge the confusion caused by our initial description of the capabilities of SGS in terms of trajectory analysis, regulatory networks, and cell-to-cell interactions. To address this, we have revised the manuscript to accurately reflect the current scope of SGS's functionalities. Specifically, SGS facilitates the visualization of pseudo time scores on UMAP scatter plots, which aids users in deducing cellular differentiation trajectories. Furthermore, by presenting peak-CRE (cis-regulatory elements and their co-accessibility) and gene expression on UMAP plots, SGS enables users to identify the intricacies of gene regulatory networks. However, it is important to clarify that SGS does not currently support the visualization of cell-to-cell communication, microenvironmental spatial organization, or clonal dynamics.

We have refined the manuscript to more accurately depict the potential value of SGS in the context of single-cell and spatial multi-omics research. The revised text now states: "**SGS plays a crucial role in advancing our understanding of cell type differentiation, the underlying gene regulatory networks, and spatial heterogeneity.**"

We believe these changes more accurately convey the visualization capabilities and potential application values of SGS without overstatement. We are grateful for your guidance, which has helped us to present our manuscript with precision and accuracy.

Other minor modification suggestions are listed below:

In Abstract/Summary, multimodal -> multimodal

Response: Thank you for your observation. We have verified the text and confirmed that the term "multimodal" is indeed used correctly in the Abstract/Summary.

Figure 1C, sCompare -> scCompare

Response: Thank you for your insightful comments. We have updated "sCompare" to "scCompare" in Figure 1C as suggested.

Page 5, Supplementary Figure 8 -> Supplementary Figure 9

Response: Thank you for your attention to detail. We have carefully reviewed and made the necessary corrections to ensure that all correspondences in the supplementary tables of the manuscript are accurate as per your suggestion.

Page 18, The back-end using Flask (version 3.02) framework, is famous for its lightweight -> The back-end uses Flask (version 3.02) framework, which is famous for its lightweight

Response: Thank you for your suggestion. We have revised the sentence on page 18: "The back-end uses Flask (version 3.02) framework, which is famous for its lightweight nature."

Page 22, To compatible with the output of these tools -> To be compatible with the output of these tools.

Response: Thank you for your valuable feedback. We appreciate your meticulous review and have made the necessary grammatical correction on page 22. The phrase has been amended: "To be compatible with the output of these tools."

Reviewer #3:

Summarize:

The authors developed a new browser called SGS, which can visualize large-scale, high-dimensional and complex data in single-cell and spatial omics, and provides functions such as multi-panel to visualize and interact with multimodal data. It is valuable for researchers as it simplifies the visualization of complex data, making it accessible to those with limited programming skills.

Minor issues:

1. Paragraph 3 in Introduction section: Authors mentioned that "SGS supports various data formats including AnnData/AnnData.zarr, MuData/MuData.zarr". The software is very friendly to researchers without programming skills. However, compared with the data formats supported by genomic data, the data formats supported by single-cell and spatial group data are relatively few. For example, data downloaded from public databases may contain data in other formats such as rds, gef, and h5, which cannot be easily loaded using SGS software.

Response: We greatly appreciate your valuable feedback highlighting the significance of accommodating diverse data formats to enhance user-friendliness, particularly for researchers lacking programming expertise.

Given that SGS is developed within a Python framework, integrating an R environment to enable the direct visualization of rds files would substantially inflate the software's size, effectively doubling it. Consequently, we recommend leveraging the `SeuratToAnndata` command from our custom-developed `SgsAnnData` R package (Figure 11A). This approach allows users to seamlessly convert rds files into the h5ad format directly within their analysis environment before uploading them to SGS for visualization, as illustrated in Figure 11B.

Furthermore, we have updated the SGS documentation to provide comprehensive guidelines on converting the `gef` file format into the h5ad format used for SGS visualization (Figure 11B). These resolutions ensure that users can effortlessly import outputs from various tools into SGS for visual exploration. Looking ahead, we plan to develop a dedicated interface conversion tool in the future to handle various single-cell and spatial multiomics data file formats, making the process even more convenient and efficient.

We hope that these enhancements can address your concerns and improve the accessibility and practicality of SGS for a wider range of users.

2.

Figure 11. (A) SgsAnnData of SGS used for the conversion of single-cell analysis objects from tools like Seurat, Giotto, Signac, and ArchR into the AnnData format. (B) Screenshots of the help documentation for the SeuratToAnndata function and additional help documentation for GEF file format conversion.

Paragraph 1 in Results section: Authors mentioned that "This makes SGS compatible with multiple systems, including Linux, Windows, MacOS, and Android." In the download link (<https://sgs.bioinfotoolkits.net/home>), the Android platform tool is not available, and Android platform is not mentioned again in the article.

Response: Thank you for your keen observation regarding the compatibility description of SGS in our manuscript. We acknowledge the discrepancy between the mention of Android compatibility and the absence of an Android platform tool in the provided download link. In response to your feedback, we have reassessed our description and decided to remove all references to the Android platform from the article. Our Android version is still under development and not yet ready for public distribution. We apologize for any confusion caused and are committed to ensuring that all statements in the article align with the resources and tools we provide.

We appreciate your professionalism and attention to detail. We will take extra care to avoid such oversights in the future and ensure the accuracy and consistency of our scientific communications.

References

1. Heffel, M.G., Zhou, J., Zhang, Y., Lee, D.-S., Hou, K., Alonso, O.P., Abuhanna, K., Schmitt, A.D., Li, T., and Haeussler, M. (2022). Epigenomic and chromosomal architectural reconfiguration in developing human frontal cortex and hippocampus. *bioRxiv : the preprint server for biology*, 2022.2010.2007.511350.
2. Goh, W.W.B., Yong, C.H., and Wong, L. (2022). Are batch effects still relevant in the age of big data? *Trends Biotechnol* 40, 1029-1040. 10.1016/j.tibtech.2022.02.005.
3. Čuklina, J., Lee, C.H., Williams, E.G., Sajic, T., Collins, B.C., Rodríguez Martínez, M., Sharma, V.S., Wendt, F., Goetze, S., Keele, G.R., et al. (2021). Diagnostics and correction of batch effects in large-scale proteomic studies: a tutorial. *Mol Syst Biol* 17, e10240. 10.15252/msb.202110240.
4. Goh, W.W.B., Wang, W., and Wong, L. (2017). Why Batch Effects Matter in Omics Data, and How to Avoid Them. *Trends Biotechnol* 35, 498-507. 10.1016/j.tibtech.2017.02.012.
5. Lazar, C., Meganck, S., Taminau, J., Steenhoff, D., Coletta, A., Molter, C., Weiss-Solís, D.Y., Duque, R., Bersini, H., and Nowé, A. (2013). Batch effect removal methods for microarray gene expression data integration: a survey. *Briefings in bioinformatics* 14, 469-490. 10.1093/bib/bbs037.
6. Ugidos, M., Nueda, M.J., Prats-Montalbán, J.M., Ferrer, A., Conesa, A., and Tarazona, S. (2022). MultiBaC: an R package to remove batch effects in multi-omic experiments. *Bioinformatics (Oxford, England)* 38, 2657-2658. 10.1093/bioinformatics/btac132.
7. Hao, Y., Hao, S., Andersen-Nissen, E., Mauck, W.M., Zheng, S., Butler, A., Lee, M.J., Wilk, A.J., Darby, C., Zager, M., et al. (2021). Integrated analysis of multimodal single-cell data. *Cell* 184, 10.1016/j.cell.2021.04.048.
8. Ni, Z., Sun, P., Zheng, J., Wu, M., Yang, C., Cheng, M., Yin, M., Cui, C., Wang, G., Yuan, L., et al. (2022). JNK Signaling Promotes Bladder Cancer Immune Escape by Regulating METTL3-Mediated m6A Modification of PD-L1 mRNA. *Cancer Res* 82, 1789-1802. 10.1158/0008-5472.CAN-21-1323.
9. He, Y.-Y., Xie, X.-M., Zhang, H.-D., Ye, J., Gencer, S., van der Vorst, E.P.C., Döring, Y., Weber, C., Pang, X.-B., Jing, Z.-C., et al. (2021). Identification of Hypoxia Induced Metabolism Associated Genes in Pulmonary Hypertension. *Front Pharmacol* 12, 753727. 10.3389/fphar.2021.753727.

10. Adamer, M.F., Brüningk, S.C., Tejada-Arranz, A., Estermann, F., Basler, M., and Borgwardt, K. (2022). reComBat: batch-effect removal in large-scale multi-source gene-expression data integration. *Bioinform Adv* 2, vbac071. 10.1093/bioadv/vbac071.
11. Leek, J.T., and Storey, J.D. (2007). Capturing heterogeneity in gene expression studies by surrogate variable analysis. *PLoS Genet* 3, 1724-1735.
12. Risso, D., Ngai, J., Speed, T.P., and Dudoit, S. (2014). Normalization of RNA-seq data using factor analysis of control genes or samples. *Nature biotechnology* 32, 896-902. 10.1038/nbt.2931.
13. Haghverdi, L., Lun, A.T.L., Morgan, M.D., and Marioni, J.C. (2018). Batch effects in single-cell RNA-sequencing data are corrected by matching mutual nearest neighbors. *Nature biotechnology* 36, 421-427. 10.1038/nbt.4091.
14. Zou, B., Zhang, T., Zhou, R., Jiang, X., Yang, H., Jin, X., and Bai, Y. (2021). deepMNN: Deep Learning-Based Single-Cell RNA Sequencing Data Batch Correction Using Mutual Nearest Neighbors. *Front Genet* 12, 708981. 10.3389/fgene.2021.708981.
15. Li, H., Brouwer, C.R., and Luo, W. (2022). A universal deep neural network for in-depth cleaning of single-cell RNA-Seq data. *Nature communications* 13, 1901. 10.1038/s41467-022-29576-y.
16. Li, X., Wang, K., Lyu, Y., Pan, H., Zhang, J., Stambolian, D., Susztak, K., Reilly, M.P., Hu, G., and Li, M. (2020). Deep learning enables accurate clustering with batch effect removal in single-cell RNA-seq analysis. *Nature communications* 11, 2338. 10.1038/s41467-020-15851-3.
17. Lotfollahi, M., Wolf, F.A., and Theis, F.J. (2019). scGen predicts single-cell perturbation responses. *Nature methods* 16, 715-721. 10.1038/s41592-019-0494-8.
18. Lopez, R., Regier, J., Cole, M.B., Jordan, M.I., and Yosef, N. (2018). Deep generative modeling for single-cell transcriptomics. *Nature methods* 15, 1053-1058. 10.1038/s41592-018-0229-2.
19. Yu, Y., Mai, Y., Zheng, Y., and Shi, L. (2024). Assessing and mitigating batch effects in large-scale omics studies. *Genome biology* 25, 254. 10.1186/s13059-024-03401-9.
20. Zhang, Z., Mathew, D., Lim, T.L., Mason, K., Martinez, C.M., Huang, S., Wherry, E.J., Susztak, K., Minn, A.J., Ma, Z., and Zhang, N.R. (2024). Recovery of biological signals lost in single-cell batch integration with CellANOVA. *Nature biotechnology*. 10.1038/s41587-024-02463-1.
21. Lin, Y., Cao, Y., Willie, E., Patrick, E., and Yang, J.Y.H. (2023). Atlas-scale single-cell multi-sample multi-condition data integration using scMerge2. *Nature communications* 14, 4272. 10.1038/s41467-023-39923-2.
22. Zhang, Z., Zhao, X., Bindra, M., Qiu, P., and Zhang, X. (2024). scDisInFact: disentangled learning for integration and prediction of multi-batch multi-condition single-cell RNA-sequencing data. *Nature communications* 15, 912. 10.1038/s41467-024-45227-w.

- 23.Zhang, Z., and Zhang, X. (2024). Data-driven batch detection enhances single-cell omics data analysis. *Cell Syst* 15, 893-894. 10.1016/j.cels.2024.09.011.
- 24.Long, Y., Ang, K.S., Sethi, R., Liao, S., Heng, Y., van Olst, L., Ye, S., Zhong, C., Xu, H., Zhang, D., et al. (2024). Deciphering spatial domains from spatial multi-omics with SpatialGlue. *Nature methods* 21, 1658-1667. 10.1038/s41592-024-02316-4.
- 25.Vandereyken, K., Sifrim, A., Thienpont, B., and Voet, T. (2023). Methods and applications for single-cell and spatial multi-omics. *Nature reviews. Genetics* 24, 494-515. 10.1038/s41576-023-00580-2.
- 26.Lopez R, N.A., Langevin M, Samaran J, Regier J, Jordan MI, Yosef N (2019). A joint model of unpaired data from scRNA-seq and spatial transcriptomics for imputing missing gene expression measurements. arXiv preprint *arXiv:1905.02269*. <https://doi.org/10.48550/arXiv.1905.02269>.
- 27.Moncada, R., Barkley, D., Wagner, F., Chiodin, M., Devlin, J.C., Baron, M., Hajdu, C.H., Simeone, D.M., and Yanai, I. (2020). Integrating microarray-based spatial transcriptomics and single-cell RNA-seq reveals tissue architecture in pancreatic ductal adenocarcinomas. *Nature biotechnology* 38, 333-342. 10.1038/s41587-019-0392-8.
- 28.Biancalani, T., Scalia, G., Buffoni, L., Avasthi, R., Lu, Z., Sanger, A., Tokcan, N., Vanderburg, C.R., Segerstolpe, Å., Zhang, M., et al. (2021). Deep learning and alignment of spatially resolved single-cell transcriptomes with Tangram. *Nature methods* 18, 1352-1362. 10.1038/s41592-021-01264-7.
- 29.Chen, A., Liao, S., Cheng, M., Ma, K., Wu, L., Lai, Y., Qiu, X., Yang, J., Xu, J., Hao, S., et al. (2022). Spatiotemporal transcriptomic atlas of mouse organogenesis using DNA nanoball-patterned arrays. *Cell* 185. 10.1016/j.cell.2022.04.003.
- 30.Moffitt, J.R., and Zhuang, X. (2016). RNA Imaging with Multiplexed Error-Robust Fluorescence In Situ Hybridization (MERFISH). *Methods Enzymol* 572. 10.1016/bs.mie.2016.03.020.
- 31.Shah, S., Lubeck, E., Zhou, W., and Cai, L. (2016). In Situ Transcription Profiling of Single Cells Reveals Spatial Organization of Cells in the Mouse Hippocampus. *Neuron* 92, 342-357. 10.1016/j.neuron.2016.10.001.
- 32.Yan, C., Zhu, Y., Chen, M., Yang, K., Cui, F., Zou, Q., and Zhang, Z. (2024). Integration tools for scRNA-seq data and spatial transcriptomics sequencing data. *Brief Funct Genomics* 23, 295-302. 10.1093/bfpg/elae002.
- 33.Kleshchevnikov, V., Shmatko, A., Dann, E., Aivazidis, A., King, H.W., Li, T., Elmentaite, R., Lomakin, A., Kedlian, V., Gayoso, A., et al. (2022). Cell2location maps fine-grained cell types in spatial transcriptomics. *Nature biotechnology* 40, 661-671. 10.1038/s41587-021-01139-4.
- 34.Andersson, A., Bergenstråhle, J., Asp, M., Bergenstråhle, L., Jurek, A., Fernández Navarro, J., and Lundeberg, J. (2020). Single-cell and spatial transcriptomics enables probabilistic inference of cell type topography. *Commun Biol* 3, 565. 10.1038/s42003-020-01247-y.
- 35.Welch, J.D., Kozareva, V., Ferreira, A., Vanderburg, C., Martin, C., and Macosko, E.Z. (2019). Single-Cell Multi-omic Integration Compares and Contrasts Features of Brain Cell Identity. *Cell* 177. 10.1016/j.cell.2019.05.006.

36. Shengquan, C., Boheng, Z., Xiaoyang, C., Xuegong, Z., and Rui, J. (2021). stPlus: a reference-based method for the accurate enhancement of spatial transcriptomics. *Bioinformatics (Oxford, England)* 37, i299-i307. 10.1093/bioinformatics/btab298.
37. Zhang, Q., Jiang, S., Schroeder, A., Hu, J., Li, K., Zhang, B., Dai, D., Lee, E.B., Xiao, R., and Li, M. (2023). Leveraging spatial transcriptomics data to recover cell locations in single-cell RNA-seq with CeLEry. *Nature communications* 14, 4050. 10.1038/s41467-023-39895-3.
38. Hu, Y., Wan, S., Luo, Y., Li, Y., Wu, T., Deng, W., Jiang, C., Jiang, S., Zhang, Y., Liu, N., et al. (2024). Benchmarking algorithms for single-cell multi-omics prediction and integration. *Nature methods* 21, 2182-2194. 10.1038/s41592-024-02429-w.
39. Satija, R., Farrell, J.A., Gennert, D., Schier, A.F., and Regev, A. (2015). Spatial reconstruction of single-cell gene expression data. *Nature biotechnology* 33, 495-502. 10.1038/nbt.3192.
40. Cheng, M., Li, Z., and Costa, I.G. (2022). MOJITOO: a fast and universal method for integration of multimodal single-cell data. *Bioinformatics (Oxford, England)* 38, i282-i289. 10.1093/bioinformatics/btac220.
41. Jin, S., Zhang, L., and Nie, Q. (2020). scAI: an unsupervised approach for the integrative analysis of parallel single-cell transcriptomic and epigenomic profiles. *Genome biology* 21, 25. 10.1186/s13059-020-1932-8.
42. Gayoso, A., Steier, Z., Lopez, R., Regier, J., Nazor, K.L., Streets, A., and Yosef, N. (2021). Joint probabilistic modeling of single-cell multi-omic data with totalVI. *Nature methods* 18, 272-282. 10.1038/s41592-020-01050-x.
43. Kriebel, A.R., and Welch, J.D. (2022). UINMF performs mosaic integration of single-cell multi-omic datasets using nonnegative matrix factorization. *Nature communications* 13, 780. 10.1038/s41467-022-28431-4.
44. Lotfollahi, M., Naghipourfar, M., Luecken, M.D., Khajavi, M., Büttner, M., Wagenstetter, M., Avsec, Ž., Gayoso, A., Yosef, N., Interlandi, M., et al. (2022). Mapping single-cell data to reference atlases by transfer learning. *Nature biotechnology* 40, 121-130. 10.1038/s41587-021-01001-7.
-

Referees' report, second round of review

Reviewer #1:

I appreciate the reviewer for their thoughtful revision. However, I still have some concerns that need to be addressed. These points are critical for readers to understand the technical advancements of SGS over existing methods. Without fully addressing them, the novelty of this study remains largely unclear.

Comparison with Vitessce: Vitessce is a major competitor of SGS. However, the differences between SGS and Vitessce are still not clearly outlined. I strongly recommend highlighting what SGS can accomplish that Vitessce cannot in each real data example presented in the "Results" section. This would better demonstrate the unique advantages of SGS.

Interactive Functionality: The authors did not seem to fully address my previous comment regarding the following: "It would be much more useful if users could select specific cells in the SC module and display the epigenomic information for those selected cells in the SG module." This functionality has already been implemented by existing visualization tools, such as Loupe Browser. The absence of this feature is a significant weakness of SGS.

Comparison with Alternative Approaches: Related to point 2, the authors' response suggests that the current version of SGS offers little advantage over the alternative scenario where "users simply open two visualizers side-by-side (e.g., Loupe Browser for single-cell data and UCSC Genome Browser for epigenomic data)." The comparison between SGS and using two visualization tools side-by-side should be emphasized in all applicable real data examples in the "Results" section. This will help clarify the added value of SGS.

Reviewer #2:

[No Comments]

Authors' response to the second round of review

RESPONSE TO REVIEWER COMMENTS:

We thank the Reviewers for the time and effort spent carefully reviewing our manuscript and providing constructive comments. Point-by-point responses to all comments and modifications are listed below. The reviewers' comments are in plain text, and our responses are in blue.

Comparison with Vitessce: Vitessce is a major competitor of SGS. However, the differences between SGS and Vitessce are still not clearly outlined. I strongly recommend highlighting what SGS can accomplish that Vitessce cannot in each real data example presented in the "Results" section. This would better demonstrate the unique advantages of SGS.

Response: Thank you very much for your constructive feedback on our manuscript. We greatly appreciate your suggestion regarding the comparison of SGS and Vitessce using practical data examples in the "Results" section. In response to your comments, Firstly, we have provided a detailed comparison of SGS and Vitessce using multiple real data examples in our reply. We have included screenshots to illustrate the differences between the two tools and provided corresponding access links for exploring. Secondly, we have thoroughly revised and supplement our manuscript, particularly in the "Results" section. For each real data example, we have also explicitly highlight the unique advantages of SGS over Vitessce, as well as the benefits of SGS compared to using multiple independent visualization tools separately, as mentioned in Comment 3. Due to the limitations of the main text length, most of the detailed comparison content have placed in an attachment related to the "Results" section. We appreciate your concerns regarding the core advantages of SGS over Vitessce, both of which aim to address the integration and visualization of single-cell and spatial multi-omics data. However, significant differences exist between the two tools in terms of development philosophy, usage patterns, functional focus, and user-friendliness. Beyond the advantages already demonstrated below in the real-data examples—such as enhanced visualization of single-cell and spatial epigenomic multi-omics data, coordinated visualization between single-cell and genome panels, 3D SRT data visualization, and collaborative data exploration—SGS's most notable strength lies in its ease of use.

Vitessce is a web-based framework that partially mitigates server dependency but limits data retrieval queries and fast navigation. It is primarily designed for users with a programming background, allowing for customized visualization portals through configuration files. However, this reliance on configuration files to customize view types, interface settings (e.g., color schemes, image transparency, and cell label display), and data loading creates a barrier to entry. It requires users to understand specific configuration parameters, reducing usability and user-friendliness (**Figure 1B**).

In contrast, SGS is a highly user-friendly, cross-platform GUI visualization software that offers rich graphical operation capabilities. Inspired by tools like Xshell and FinalShell, SGS leverages Docker and Flutter technologies and adopts a client-server separation strategy to provide a rapid and user-friendly platform for non-programmers, emphasizing team collaboration and real-time data sharing (video demo link). SGS overcomes complex configurations and environmental dependencies while ensuring compatibility across Linux, Windows, and macOS. Users can perform operations such as graphical one-click installation, theme setting, single-cell configuration, and track style settings through a GUI without needing to edit configuration files directly (**Figure 1A**).

Figure 1 . Comparative Analysis of Installation and Interface Usability between SGS and Vitesce . (A) SGS features a streamlined graphical installation process and user -friendly interface operations, designed to enhance ease of use and accessibility. (B) Vitesce , in contrast, involves a more complex configuration process and requires detailed interface settings, reflecting a more technical approach to setup and usability.

Below, we provide a detailed discussion of the differences and advantages of SGS compared to Vitesce using real data examples from the “Results” section.

Example 1: PBMCs scATAC-seq 5K Data Visualization

To provide a more intuitive comparison of the visualization capabilities between Vitesce and SGS, we utilized the same dataset (scATAC-seq data of peripheral blood mononuclear cells from 10X Genomics) and conducted demonstrations on both Vitesce and SGS. The demonstration links are as follows: SGS (demo link) and Vitesce (demo link). Our comparison revealed that while both tools support basic data visualization functions such as single-cell clustering map rendering, SGS significantly outperforms Vitesce in the integration and visualization of snATAC data.

Firstly, the multi-panel collaborative design of SGS is a major highlight, especially in the integration and visualization of snATAC multimodal data. For instance, SGS allows users to mark peaks within the single-cell panel and then quickly examine the expression distribution differences of these peaks across different

cell types, while simultaneously navigating to the associated genomic regions (**marked as 1 and 5 in Figure 2**). This feature enables users to delve into the chromatin accessibility signal distribution among various cell types. In contrast, Vitesse does not support the association and navigation between marked peaks and epigenetic track signals.

Secondly, in response to the second comment from Reviewer 1, we further optimized the panel collaboration functions of SGS. Users can select specific cells in the SC module, and SGS will immediately display the corresponding epigenomic information in the SG module (**marked as 2 and 4 in Figure 2**). This collaborative functionality greatly enhances the practical value of SGS, as it not only eliminates the cumbersome need to open two separate visualization tools but also achieves seamless integration of scATAC data. In comparison, Vitesse lacks this key function, resulting in weaker integration of multimodal data.

Lastly, in terms of data visualization content, SGS supports the visualization of gene-CRE (gene-cis-regulatory element) links, which is crucial for elucidating gene regulatory networks and mechanisms of gene expression regulation. SGS intuitively represents the interaction strength differences between various sites using curves of varying opacity, helping users easily identify key gene-CRE links (**marked as 3 in Figure 2**). Vitesse, on the other hand, lacks the ability to visualize gene-CRE links, limiting its application in exploring gene regulatory mechanisms.

In summary, SGS's superiority in gene-CRE links visualization (**marked as 3 in Figure 2**), multi-panel coordination (**marked as 2 and 4 in Figure 2**), and rapid navigation functions (**marked as 1 and 5 in Figure 2**) significantly enhances its ability to integrate and visualize scATAC-seq data compared to Vitesse.

Figure 2. (A) Screenshot of SGS visualization for pbmc5k scATAC data. (B) Screenshot of Vitesce visualization for pbmc5k scATAC data.

Example 2: Single-nucleus Methyl-3C Sequencing (sn-m3C-seq) Data Visualization Single-nucleus methyl-3C sequencing (sn-m3C-seq) is a high-throughput sequencing technology that enables the simultaneous detection of DNA methylation and 3D chromatin conformation within individual cell nuclei, revealing the interplay between genomic structure and epigenetic modifications. To demonstrate the unique value of SGS, we used the same dataset (Human PFC and HPC sn3C-seq epigenetic multimodal data) to conduct demonstrations on both SGS and Vitesce [1]. The demonstration links are as follows: SGS (demo link) and Vitesce (demo link). SGS can integrate complex sn-m3C-seq epigenetic multimodal data into a

unified interface for visualization, offering significant functional advantages over Vitesce, as detailed below.

Firstly, SGS supports the display of cell clustering alongside genome-mapped tracks such as CG methylation signals and HiC tracks for specific cell types (**marked as 2 in Figure 3A**). This integrated display allows users to efficiently explore the relationship between chromatin interactions and DNA methylation in specific cell populations. In contrast, Vitesce currently does not support the visualization of HiC chromatin interactions and methylation intensity data, and thus cannot display such complex sn-m3C-seq epigenomic multimodal data in a single interface.

Secondly, SGS, however, goes beyond this by offering a multi-panel coordination design and adaptive communication mechanism that enhances the sn-m3C-seq dataset exploration. For example, users can click on specific features in the single-cell marker table (such as the *RORB* gene) to quickly navigate to the corresponding genomic region and obtain multi-level information, including chromatin accessibility, HiC interaction intensity, and methylation signal distribution (**marked as 1 and 3 in Figure 3A**). This collaborative functionality greatly improves the efficiency of data exploration. However, Vitesce currently does not support the co-visualization of these epigenomic genome-mapped signals with single-cell atlas, nor does it support rapid localization of marked features. Users must manually zoom in step-by-step to navigate to features of interest.

Lastly, SGS provides advanced visualization features, such as support for multiple HiC data normalization methods (e.g., KR, VC, VC_SQRT) and the ability to set custom interaction intensity thresholds for data filtering. In comparison, Vitesce's components currently do not support the visualization of complex epigenomic data such as HiC and methylation, and thus cannot offer data normalization and filtering functions.

In summary, this case highlights the unique value of SGS in visualizing sn-m3C-seq epigenomic multimodal data. Through its multi-panel integration, adaptive communication functions (**marked as 1 and 3 in Figure 3A**) and advanced genome browser (**marked as 2 in Figure 3A**), SGS enables the integrated visualization of epigenome genome-mapped signals (gene annotation, DNA methylation, 3D chromatin conformation) and single-cell methylation information within a unified interface, offering insights that are not achievable with Vitesce.

Example 3: Single-cell eQTL Demo Data Visualization

Single-cell expression quantitative trait loci (sc-eQTL) refer to the associations between genetic variations (such as SNPs) and gene expression at the single-cell level. These associations typically occur in specific gene regulatory regions, such as enhancers or promoters, which play a crucial role in cell type specificity, dynamic changes, and responses to external stimuli. In this study, we used the sc-eQTL multimodal dataset

from the OneK1K project to intuitively compare the functional differences between SGS (demo link) and Vitessce (demo link), highlighting SGS's strengths in visualizing sc-eQTL multimodal data [2].

Firstly, SGS synchronizes single-cell panels with the genome browser panel using sc-eGenes (genes significantly associated with sc-eQTL) as anchors, efficiently identifying sc-eQTL regions related to cell type-specific expression (**marked as 2 and 3 in Figure 4A**). For example, users can click on a marked gene (such as *BLK*) in the single-cell panel to intuitively display its heterogeneous expression patterns across different cell types via the single-cell expression map. Simultaneously, SGS can quickly navigate to the sc-eQTL region significantly associated with *BLK* and highlight the sc-eQTL loci regulating *BLK* expression in red based on significance. In contrast, Vitessce cannot display gene-associated regions of sc-eQTL epigenetic multimodal data, limiting its ability to explore the relationship between genetic variation and cell function specificity.

Secondly, SGS also offers robust genome navigation and query functions, enabling users to easily locate specific sc-eQTL regions by entering gene symbols (e.g., *BLK*) or genomic positions (e.g., chr8:11300703-11611527) and view detailed information such as p values and site IDs. Additionally, users can set specific thresholds based on P-values, and SGS will rapidly filter out the sc-eQTL regions that meet these criteria (**marked as 1 in Figure 4A**). These advanced visualization features significantly enhance user experience and work efficiency. In comparison, Vitessce lacks these functions, requiring users to manually perform zooming, dragging, and data filtering operations, which greatly impacts user experience. Overall, SGS's advantages in displaying associations between gene expression and genetic variation (**marked as 2 and 3 in Figure 4A**) and in data filtering and navigation (**marked as 1 in Figure 4A**) make it a better choice than Vitessce in visualizing sc-eQTL multimodal data. These features not only improve the efficiency of data exploration but also provide researchers with more intuitive and convenient tools to deeply investigate the regulatory mechanisms of gene expression by genetic variations.

Figure 4. (A) The screenshot of OneK1K sc-eQTL visualization demo of SGS. **(B)** The visualization of sc-eQTL multimodal data in Vitesce.

Example 4: 3D Spatial Transcriptome Demo Data Visualization

Compared to traditional 2D slices, organ-level and organism-level 3D spatially resolved transcriptomics (SRT) offer a more comprehensive understanding of biological principles. In this context, SGS significantly outperforms its primary competitor, Vitesce, in the visualization of 3D SRT data. This study highlights the distinct advantages of SGS (demo link) using a 3D SRT dataset from *Drosophila melanogaster* embryos

(16–18 hours after egg laying) as an example [3]. Vitessce currently lacks support for visualizing 3D spatially resolved transcriptomics datasets. Therefore, in the following demonstrations, we will exclusively use SGS to present data examples, thereby fully highlighting its visualization capabilities.

In terms of data visualization support, Vitessce is unable to visualize 3D SRT expression data and mesh surface models. In contrast, SGS excels in visualizing 3D SRT data based on continuous 2D virtual slice reconstruction. It can clearly display different regions of the fruit fly embryo, such as the central nervous system, midgut, fat body, and epidermis. Additionally, SGS supports rapid rendering of mesh surface models, as well as functions like zooming, rotating, and switching between groups (**Figure 5a**). This enables users to explore the 3D SRT data along different body axes or from various angles of complex organs. However, Vitessce lacks the capability to display mesh surface models.

In capturing spatial expression patterns of specific regions, SGS allows users to search for marker genes unique to each tissue and observe spatial expression pattern changes in the endoderm, mesoderm, and ectoderm regions (**Figure 5b**). For instance, *Try29F* (a trypsin family gene) shows specific expression in the posterior midgut, providing important clues for studying the functional regionalization of the midgut during embryogenesis. Vitessce, due to its lack of relevant functional support, is unable to conduct in-depth visualization of such specific regional gene expression patterns.

Finally, Users can also visualize multiple genes (such as *Rbp6* in the CNS, *grh* in the epidermis, *srp* in the fat body, *kay* in the midgut, and *Mef2* in muscles, etc) of interest simultaneously with the gene expression module of SGS (**Figure 5c**).

In summary, SGS offers advanced 3D transcriptomic data visualization capabilities that significantly enhance interactive 3D data exploration. Unlike Vitessce, SGS supports detailed surface model visualization, animate the mesh models, view the expression heterogeneity in 3D context, and multi-gene visualization. These features bridge the gap between researchers and spatially resolved transcriptomics, making SGS a powerful tool for studying complex biological processes such as embryogenesis and organogenesis.

Figure 5. The 3D SRT dataset of late-stage Drosophila embryos (16–18 hours post-oviposition) is displayed through the SGS browser. The left panel presents the surface model of the Drosophila embryo tissue, such as CNS, midgut, fat body, and epidermis (a). The right panel reveals the spatial expression distribution of *Try29F* (b). The bottom panel displays the spatial feature maps of marker genes such as *Mal-A3*, *Try29F*, *Rbp6*, *Mef2*, *kay*, and *X11L* (c).

Example 5: Genome-mapped Demo Data Visualization

Comprehensive visualization of genome-mapped data is essential for exploring complex single cell and spatial epigenomic data, and SGS demonstrates significant advantages over Vitessce in this regard.

In the genome-mapped data demo visualization, unlike Vitessce, which primarily supports basic genome-mapped data types such as chromatin accessibility signals and gene structure, SGS's genome browser offers a rich array of track visualizations (demo link), including gene structure, genome variation loci, and epigenetic signals (such as chromatin accessibility signals, differential peaks, gene-CRE links, methylation signals, chromatin interactions, and sc-eQTL loci information), and more (as displayed in **Figure 6A**).

Moreover, SGS also provides advanced genome track operations such as feature/region location, data scaling, filtering, statistical analysis, detailed information display, and flexible track splitting and grouping settings. These capabilities ensure that users can deeply explore and analyze data in detail according to their needs. In contrast, Vitessce's genome framework is limited to basic operations like zooming in/out and lacks essential retrieval functions, such as querying by location or gene name.

Furthermore, SGS's dual-chromosome visualization module provides a powerful visual comparison of single-cell and spatial epigenomic multimodal signals (video demo link) (**Figure 6B**). While the genome framework of Vitessce can do little of that. By utilizing a double chromosome display strategy, this module

of SGS presents several advantages. Firstly, the top and bottom coordinates can cover different genomic regions, enabling the visualization of long-range interactions. Secondly, users have the flexibility to independently shift or zoom these coordinate regions. This capability facilitates the comparative visualization of epigenomic signal differences in multiple regions simultaneously, enhancing our understanding of regulatory patterns.

Overall, the novel, flexible, and scalable genome browser framework of SGS offers a scalable solution for integrating an ever-expanding volume of epigenomic multimodal genome-mapped datasets.

Figure 6. (A) The screenshot of genome -mapped data visualization. The panel is displayed from top to bottom gene structure, genome variation loci, and chromatin interactions etc. (B) The SGS double -chromosome visualization mode of SGS. The top and bottom panels display bigwig track and Hic interaction track on chr1 and chr7, respectively. The Big interact track in the center panel shows interaction derived links between the genomic regions (top coordinate) and other genomic regions (bottom coordinate). The strengths of the genomic interactions are plotted in color scale, with red being strongest.

Example 6: Collaborative Data Visualization

Team collaboration is essential for the exploration of single-cell and spatial multi-omics visualization, and SGS significantly outperforms existing tools like Vitessce in this critical area. SGS provides a robust framework for collaborative data exploration through its comprehensive user management system, data management system, and advanced collaborative visualization features (video demo link). In contrast, Vitessce currently lacks support for multi-user collaboration and data management features, which significantly limits its applicability in team-based research settings. As a result, we focus solely on SGS for the following data demonstrations to fully highlight its strengths in collaborative visualization.

Firstly, in the aspect of data collaborative visualization, SGS excels in collaborative visualization by enabling real-time, multi-user collaboration and editing, document commenting, cell type annotation, access management, and data management. These functionalities are not available in Vitessce. Specifically, **Figure 7A** illustrates how multiple users can collaboratively visualize genome-mapped data. Users can select specific features (gene or peak) to rename them, add relevant comments, and review the collaborative operation history. By clicking on a record in the history list, SGS swiftly locates the position of the feature modification, ensuring efficient and transparent collaboration. Additionally, **Figure 7B** highlights the single-cell collaborative visualization features. SGS supports collaborative annotation by allowing multiple users to annotate the same dataset. Its dual-screen mode displays the annotation results from different users side-by-side, thereby facilitating efficient team collaboration in single-cell projects.

Secondly, in the aspect of user and data management, SGS supports the addition of multiple users and the assignment of different user permissions, enabling seamless collaboration within a shared visualization environment. Once users are added, they can log into the same visualization service to perform collaborative data visualization operations and manage project data. These capabilities are notably absent in Vitessce.

Overall, SGS is designed to empower research teams to rapidly establish a visualization and sharing platform for collaborative data exploration, co-annotation, commenting, and project data/user management—without the need for programming. SGS's advanced collaborative features offer a substantial improvement over Vitessce, making it a better choice for team researchers engaged in single-cell and spatial multi-omics collaborative visualization.

Figure 7. Collaborative Visualization Features of SGS. (**A**) Genome-mapped Data Collaboration. SGS supports multiple users in adding comments to features (**a**), viewing collaboration history (**b**), and renaming features (**c**) for collaborative visualization. (**B**) Collaborative Cell Annotation. SGS enables multiple users to create annotation groups (**a**); and compare different users' annotation results, such as results from User 1 and User 2 (**b**, **c**).

Finally, to address the reviewers' concerns, we have thoroughly revised and expanded various sections of the manuscript, as well as the supplementary materials. The specific revisions are outlined below:

(1) Introduction Section: In the overview and limitations of existing visualization tools in the introduction, we further supplement the limitations of using multiple independent visualization tools side-by-side to achieve multimodal data visualization (line 63-66).

(2) Overview of SGS Section: We have refined the descriptions of SGS's capabilities (lines 95-120) and provided detailed supplementary notes (supplementary lines 1-300) to support these revisions. First, we conducted a detailed comparison between SGS and the approach of using multiple independent tools side-by-side for multimodal data visualization (supplementary lines 14-53), and then we provided an in-depth comparison between SGS and Vitessce, using the example datasets from the Results section to demonstrate SGS's superior performance in complex multimodal data visualization (supplementary lines 54-300).

(3) Enhanced User Friendliness via Graphical Installation and Interface Ease Section Added: We have expanded this section to elaborate on the user-friendliness of SGS in terms of installation and interface interactivity. Additionally, we have included a comparative analysis with Vitessce regarding installation and interface configuration (line 176-197; Figure S5-S6).

(4) Multi-user Collaborative Data Visualization and Management Section Added: We have introduced a new section in the revised manuscript to better highlight the advantages of SGS in multi-user collaborative data visualization (line 199-219). Additionally, we have provided supplementary information in the appendix regarding collaborative visualization of single-cell and genome-mapped data, as well as user and data management (Figure S7-S8).

(5) User Case Section: In this section, we have expanded the case studies of sn-m3C-seq and sc-eQTL to further illustrate the unique advantages of SGS over Vitessce in visualizing these multimodal datasets (Figure S2-S3).

We hope these comprehensive revisions and additions can address your concerns by providing a clearer and more detailed presentation of SGS's capabilities and advantages.

Interactive Functionality: The authors did not seem to fully address my previous comment regarding the following: "It would be much more useful if users could select specific cells in the SC module and display the epigenomic information for those selected cells in the SG module." This functionality has already been implemented by existing visualization tools, such as Loupe Browser. The absence of this feature is a significant weakness of SGS.

Response: Thank you very much for raising this important point. After a thorough re-examination, we have confirmed that Loupe Browser does indeed support the selection of specific cell clusters in the single-cell panel and the real-time display of corresponding epigenomic information in the genomic browser panel. However, it should be noted that Loupe Browser fundamentally differs from mainstream single-cell ATAC

analysis tools (such as ArchR and SnapATAC) in how it calculates chromatin accessibility differences between cell clusters. Loupe Browser indirectly reflects chromatin accessibility differences by comparing the proportion of cells with chromatin accessibility scores within genomic regions. In contrast, SGS previously adopted a strategy of directly visualizing the true coverage scores of each cell type at genomic positions, which aligns with the statistical approaches used by tools like ArchR and SnapATAC. We will elaborate on the advantages and disadvantages of these two strategies in the following sections.

To further enhance the interactivity between the single-cell panel (SC panel) and the genomic browser panel (SG panel) in SGS, we have adopted suggestion 2 from Reviewer 1 and incorporated visualization strategies inspired by Loupe Browser. In the latest version of SGS, we have added a collaborative visualization feature that allows users to select specific cell clusters in the SC panel and view their corresponding epigenomic information in real-time in the SG panel. Additionally, we have provided a demonstration video (video demo link) to help users better understand how to use this new feature. Users can install and log in to the SGS software using our provided demo dataset (SGS_scATAC_pbmc5k_demo2.h5ad) to experience this new functionality firsthand. Here are the detailed analyses and explanations: **(1) Loupe Browser's logic for calculating cell-specific chromatin accessibility signals**

Loupe Browser displays peaks on genomic tracks in the form of vertical bars within cluster tracks. The width of each bar is proportional to the peak's width, and the height is proportional to the percentage of cells within that cluster track where chromatin is open at that peak. This method allows users to intuitively see the difference in the proportion of cells with chromatin accessibility in each peak across different cell clusters, which also reflects the differences in cell accessibility to some extent. The main advantage of this approach is its intuitiveness and low computational demand, making it suitable for quickly comparing the chromatin accessibility differences of predefined peak regions among cell clusters. However, it also has certain limitations. It can only examine predefined peak regions and cannot capture chromatin accessibility changes outside these regions. Moreover, it fails to display the real differences in chromatin accessibility signals among different cell clusters in specific regions. **(2) Logic for calculating chromatin accessibility signals in tools like ArchR**

ArchR primarily generates BigWig signal files for each cell cluster by splitting the fragment files of single-cell ATAC-seq data. Specifically, it creates fragment files containing all cells belonging to a specific cluster and then counts the number of Tn5 insertions in the corresponding peak set. The numbers of Tn5 insertions are normalized by the number of reads in peaks and computed in windows genome-wide using "slidingWindows (chromSizes, 100, 100)". To achieve this, the genome is divided into 100-bp intervals using "tile" in GenomicRanges based on chromosome sizes. The insertion sites (GenomicRanges) are then converted into a coverage run-length encoding using "coverage" and the sum is calculated in each bin. This method mainly calculates the chromatin accessibility signal strength for each genomic location in each cell cluster, rather than cell proportions. The advantage of this approach is that it can display chromatin accessibility signals across the whole genome, not just in predefined peak regions, offering higher flexibility and resolution. However, it also has certain limitations. The computational cost is high, as generating BigWig signals requires complex processing of fragment files, which consumes more memory and slows down the rendering speed, potentially degrading the user experience.

Overall, both Loupe Browser and ArchR have their strengths and limitations when dealing with single-cell ATAC-seq data. Loupe Browser is more suitable for quickly displaying and comparing the chromatin accessibility differences of predefined peak regions among cell clusters, while ArchR's method provides higher flexibility and resolution, allowing for the capture of a broader range of chromatin accessibility changes.

(3) Improvements in SGS

Our previous understanding of real-time generation and visualization of chromatin accessibility signals for specific cell clusters was mainly based on the approach of mainstream analysis tools like ArchR. We found that implementing this method to display the corresponding epigenomic signals for selected cell clusters in the genome browser panel involves significant computational demands, high memory consumption, and slower rendering speeds, all of which degrade the user experience. We have recognized that although SGS has certain interactive features for integrating and visualizing multimodal data, it falls short in real-time generation and display of chromatin accessibility signals for specific cell clusters. To address this deficiency, we have integrated a calculation method similar to that of Loupe Browser in the latest version of SGS. This enables users to select specific cell clusters in the single-cell panel and display their corresponding epigenomic signals in the genome browser panel in real-time. We have further enhanced the interactive functions between multiple panels in SGS, promoting integrated visualization across panels and allowing users to quickly switch and compare data between different panels.

We are extremely grateful for your valuable suggestions, which not only helped us identify the shortcomings of SGS in this regard but also spurred us to make improvements and optimizations. Finally, we added this feature in the SG section of the manuscript (line 341-346) and updated the help document to help users better experience this feature. We believe that these enhancements will make SGS more comprehensive in visualizing single-cell and spatial multimodal data and better meet users' needs.

Comparison with Alternative Approaches: Related to point 2, the authors' response suggests that the current version of SGS offers little advantage over the alternative scenario where "users simply open two visualizers side-by-side (e.g., Loupe Browser for single-cell data and UCSC Genome Browser for epigenomic data)." The comparison between SGS and using two visualization tools side-by-side should be emphasized in all applicable real data examples in the "Results" section. This will help clarify the added value of SGS.

Response: We appreciate the reviewer's insightful comments regarding the comparison between SGS and using two separate visualization tools side-by-side (such as Loupe Browser for single-cell data and UCSC Genome Browser for epigenomic data). We fully agree that emphasizing the added value of SGS in this context is crucial for demonstrating its utility. In response, we have carefully revised our manuscript to more explicitly emphasize the advantages of SGS over using multiple independent tools side-by-side in the "Results" supplementary section. Here, we have emphasized and explained the unique value of SGS here. Specifically, we have highlighted the following three key aspects:

1. Integration and Synergistic Visualization of Multimodal data

SGS excels in its ability to integrate and visualize multimodal data within a unified framework. Unlike using two separate tools side-by-side, SGS offers an integrated platform that allows for seamless visualization of both single-cell and complex epigenomic data (sn-m3C-seq, sc-eQTL data ect) through a multi-panel interactive design. SGS utilizes genome features (such as peaks, genes, and sc-eQTL) as interactive anchors to achieve coordinate visualization between the single-cell panel and the genome browser. Users can click on marker genes or peaks to simultaneously view their dynamic changes at the single-cell level and navigate to the genome browser to observe variations in epigenetic signals and provide detailed annotations of these features. For scATAC data, the latest version of SGS allows users to select specific cell clusters in the single-cell panel and display their corresponding epigenomic signals in the genome browser panel in real-time (video demo link). For single-cell eQTL data, SGS allows users to click on marker genes in the sc-eGenes table to view the gene's expression changes, while highlighting the associated sc-eQTL loci in the genome browser. This feature provides an intuitive display of the association between genetic variations and cellular functions, which is unattainable when using separate tools. Overall, these interactive designs offer a more comprehensive and efficient cross-modal integration, providing a holistic view of complex multimodal data that is not possible by using disconnect tools side-by-side.

2. Efficiency and Simplicity in Multimodal data Visualization

Compared to using multiple independent visualization tools, SGS's integrated design simplifies the exploration of complex multimodal data in the aspect of data management and Caching Capabilities

2.1 Data Management and Integration

SGS's data management system offers significant advantages over using independent visualization tools, in the areas of data upload and management. Using disconnect visualization tools face limitations when dealing with the complexity of multimodal data, which are often high-dimensional, noisy, and diverse. These tools require cumbersome individual operations for data addition, grouping, and deletion, which are inefficient and time-consuming. For example, Loupe Browser only supports its native .loupe file format, not the more widely-used anndata format. This limitation forces users to perform extensive data format conversions and repeated uploads, further complicating the visualization process. In contrast, SGS's data management system is specifically designed for efficient batch management of complex multimodal data. It enables users to effectively manage multiple projects with features for batch addition, deletion, and data grouping, thereby enhancing efficiency and flexibility. By consolidating these functions into a single platform, SGS allows users to complete the entire workflow from data import to visualization within one interface, avoiding the need to switch between multiple tools.

2.2 Advanced Data Caching Capabilities

Another significant advantage of SGS over using independent tools like Loupe Browser and UCSC Genome Browser for multimodal data visualization lies in its advanced data caching capabilities. SGS is designed to cache the visualization information of single-cell and genomic panels, allowing users to resume their previous visualization view seamlessly even after closing and reopening the software. This feature ensures that users can pick up where they left off without the need to re-upload or reconfigure data. In contrast, tools like Loupe Browser and UCSC Genome Browser do not offer such robust caching mechanisms. When

visualizing complex multimodal data such as sn-3C-seq or sc-eQTL, users must re-upload and re-operate the data in each tool separately to restore their previous visualization views. This process is not only time-consuming but also increases the risk of errors and inconsistencies.

3. Multi-user Collaborative Visualization

SGS's integrated panel design significantly enhances the collaborative experience for teams.

Compared to using independent tools for single-cell and spatial multi-omics data exploration, SGS's integrated panels and collaborative features enable multiple users to visualize and interact with genome-mapped data, single-cell data, and spatial data simultaneously. Users can perform operations such as feature renaming, track commenting, and joint cell-type annotation within a single platform. Additionally, SGS supports sharing current views via session URLs, a feature that is not available when using Loupe Browser and UCSC Genome Browser separately. This capability is essential for team collaboration and significantly enhances the efficiency of collaborative research.

Finally, we have also used the sn-m3C-seq data in the "Results" section as an example to illustrate this comparison through a video demonstration (video link). In this example, we encountered some issues with environment dependencies and format incompatibilities when converting the h5ad format data to the .loupe file. Therefore, we used cellxgene as an alternative to Loupe Browser for single-cell data visualization and UCSC Genome Browser for visualizing genome-mapped data such as methylation signals and chromatin interactions. The video demonstrates how SGS integrates these diverse data types into a unified interface, which is more efficient, user-friendly and collaborative compared to the scenario where users have to open and manually synchronize multiple visualization tools side-by-side. We hope this comparison can clearly highlight the added value of SGS in terms of multimodal data integration visualization.

In summary, SGS demonstrates significant advantages in multimodal data integration visualization, team collaboration efficiency and simplicity. These strengths enable it to better meet researchers' needs for analyzing complex multimodal datasets than using disconnected tools like Loupe browser and UCSC Genome Browser. To further clarify the added value of SGS, we have revised the manuscript (lines 63-66;343-347) and provided additional explanations in the supplementary files (lines 15-55). Thank you again for your suggestions and guidance.

References

- [1] Granja, J.M., Klemm, S., McGinnis, L.M., Kathiria, A.S., Mezger, A., Corces, M.R., Parks, B., Gars, E., Liedtke, M., Zheng, G.X.Y., et al. (2019). Single-cell multiomic analysis identifies regulatory programs in mixed-phenotype acute leukemia. *Nature biotechnology* 37, 1458-1465. [10.1038/s41587-019-0332-7](https://doi.org/10.1038/s41587-019-0332-7).
- [2] Yazar, S., Alquicira-Hernandez, J., Wing, K., Senabouth, A., Gordon, M.G., Andersen, S., Lu, Q., Rowson, A., Taylor, T.R.P., Clarke, L., et al. (2022). Single-cell eQTL mapping identifies cell type-specific genetic control of autoimmune disease. *Science (New York, N.Y.)* 376, eabf3041. [10.1126/science.abf3041](https://doi.org/10.1126/science.abf3041).
- [3] Guo, L., Li, Y., Qi, Y., Huang, Z., Han, K., Liu, X., Liu, X., Xu, M., and Fan, G. (2023). VT3D: a visualization toolbox for 3D transcriptomic data. *J Genet Genomics* 50, 713-719.

[10.1016/j.jgg.2023.04.001](https://doi.org/10.1016/j.jgg.2023.04.001).

Referees' report, third round of review

Reviewer #1:

My comments have been addressed.